# Far from the Shallow: Brain-Predictive Reasoning Embedding through Residual Disentanglement

**Linyang He**[1,2*]  **Tianjun Zhong**[3*]  **Richard Antonello**[1,2]
**Gavin Mischler**[1,2]  **Micah Goldblum**[2]  **Nima Mesgarani**[1,2]
[1]Zuckerman Mind Brain Behavior Institute, Columbia University
[2]Department of Electrical Engineering, Columbia University
[3]Department of Computer Science, Columbia University
Correspondence: linyang.he@columbia.edu  nima@ee.columbia.edu

## Abstract

Understanding how the human brain progresses from processing simple linguistic inputs to performing high-level reasoning is a fundamental challenge in neuroscience. While modern large language models (LLMs) are increasingly used to model neural responses to language, their internal representations are highly "entangled," mixing information about lexicon, syntax, meaning, and reasoning. This entanglement biases conventional brain encoding analyses toward linguistically shallow features (e.g., lexicon and syntax), making it difficult to isolate the neural substrates of cognitively deeper processes. Here, we introduce a residual disentanglement method that computationally isolates these components. By first probing an LM to identify feature-specific layers, our method iteratively regresses out lower-level representations to produce four nearly orthogonal embeddings for lexicon, syntax, meaning, and, critically, reasoning. We used these disentangled embeddings to model intracranial (ECoG) brain recordings from neurosurgical patients listening to natural speech. We show that: 1) This isolated reasoning embedding exhibits unique predictive power, accounting for variance in neural activity not explained by other linguistic features and even extending to the recruitment of visual regions beyond classical language areas. 2) The neural signature for reasoning is temporally distinct, peaking later (~350-400ms) than signals related to lexicon, syntax, and meaning, consistent with its position atop a processing hierarchy. 3) Standard, non-disentangled LLM embeddings can be misleading, as their predictive success is primarily attributable to linguistically shallow features, masking the more subtle contributions of deeper cognitive processing. Our work provides compelling neural evidence for an abstract reasoning computation during language comprehension and offers a robust framework for mapping distinct cognitive functions from artificial models to the human brain[2].

A growing body of work has shown that LLMs exhibit strong representational alignment with human brain activity during language comprehension. These studies employ *language encoding models*, powerful predictors of measured brain activity in response to a language stimulus [41, 19, 10, 20, 38, 5, 13, 34, 17, 7, 27, 2, 3, 8, 9]. Typically, these models are created by inputting a language stimulus into an LLM, to create a contextual embedding of the stimulus using the models' internal hidden state. A linear mapping is then learned between these hidden states and the measured brain response to that stimulus. Modern language encoding models represent some of the most advanced computational approaches for studying neural processing across sensory modalities, and they provide a unique window into the semantic processing landscape of human cortex.

---

[*]Equal Contribution.
[2]Code available here.

39th Conference on Neural Information Processing Systems (NeurIPS 2025).

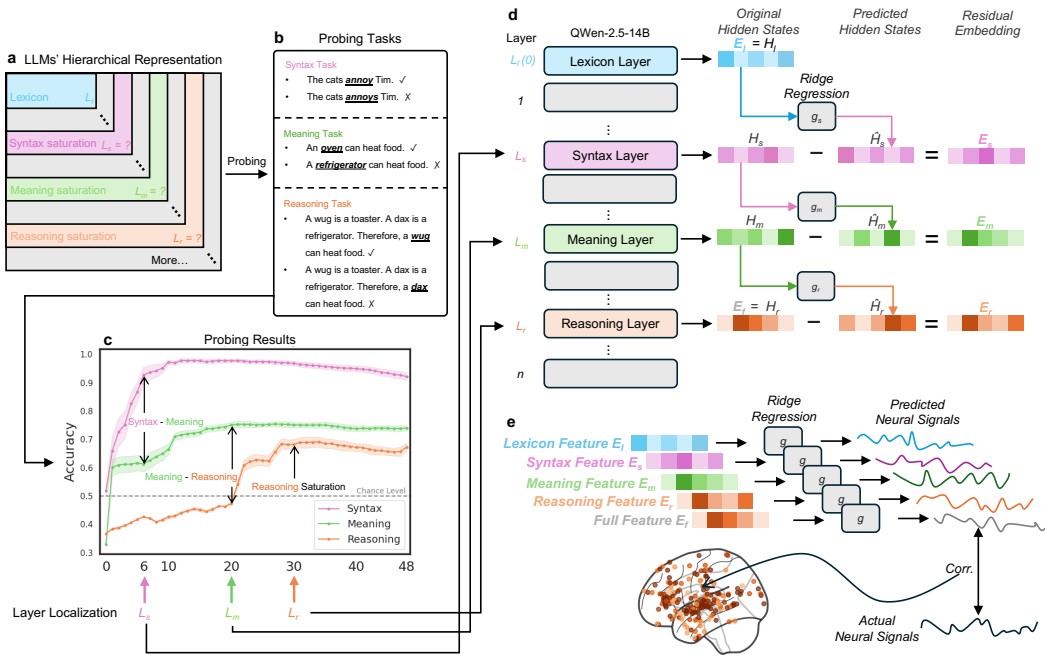

Figure 1: **a)** Hierarchical representations in an LLM. Transformer layers accumulate information in order: lexical features emerge first, followed by syntax, contextual meaning and eventually higher-order reasoning, with still-richer knowledge continuing in later layers. **b)** Minimal-pair probing tasks. Three diagnostic sentence sets separately test syntax, concept meaning and multi-premise reasoning. **c)** Layer localization from probing curves. We define $L_s$ – the earliest layer where syntax performance saturates while meaning is still low; $L_m$ – layer where meaning saturates but reasoning has not yet emerged; and $L_r$ – layer where reasoning performance plateaus. These identified layers through probing will be used in later analyses. **d)** Feature disentangling across layers. Starting from the localized layers, we iteratively regress lower-level features out of higher ones. Details of residual embedding constructions could be found in Algorithm 1. Residual disentangling yields four orthogonal embeddings that isolate lexicon, syntax, meaning and reasoning information. **e)** Brain encoding with purified features. Each residual feature is fed into a ridge encoder to predict high-gamma ECoG responses of Podcast listening. Comparing predicted and actual neural signals reveals the spatiotemporal distribution of cortical activity uniquely associated with lexicon, syntax, meaning and reasoning representations.

However, most work that has studied this brain-LLM alignment has focused on semantics and low-level phonological relationships [1, 39, 32]: little is known about the degree to which LLMs and the brain align in their higher-level reasoning processes. One reason is historical: reasoning capabilities have only recently emerged robustly in modern LLMs [21]. Another issue is methodological. Most prior studies have largely treated language encoding models as monolithic black boxes that map entire hidden states to brain responses without disentangling the specific linguistic or cognitive functions represented within them, such as lexical features, syntactic structure, semantic meaning, or reasoning processes.

Recent findings [26] show that when models are trained to perform multiple tasks of varying complexity, their internal representations disproportionately favor simpler and more linearly extractable features, even when accuracy on complex tasks is equally high. This suggests that in brain alignment studies, unsegmented LLM features may reflect a bias toward lexical or syntactic components simply because they are computationally easier to represent.

To move beyond these biases and probe the deeper relationship between machine and human reasoning, we introduce a novel approach: residual reasoning embeddings. Rather than comparing raw LLM activations to brain activations, we isolate the reasoning-specific component of LLM hidden states by separating them from discrete lexical, syntactic, and meaning features. This residual embedding

captures the unique portion of variance in the initial embedding that is attributable to high-level reasoning information.

We then use these residual reasoning embeddings to build encoding models that predict human electrocorticographic (ECoG) brain activity during tasks that require inference. We demonstrate that these reasoning representations align with distinct spatiotemporal neural patterns, peaking later in time and engaging more frontal and visual regions than lexical or syntactic signals, suggesting a shared computational hierarchy between LLMs and the human brain. Furthermore, we show that full LLM embeddings are biased toward lower-level features, and that only through disentanglement can reasoning-related brain activity be meaningfully isolated. Together, our results offer a new lens on both model interpretability and the neural basis of abstract linguistic reasoning.

## 2 Methods

### 2.1 Probing Datasets

To identify the LLM layers most specialized for encoding syntactic, semantic, and reasoning features, we apply feature-specific probing using two diagnostic datasets. For syntactic probing, we use the Benchmark of Linguistic Minimal Pairs (BLiMP) [40], which evaluates grammatical sensitivity across 67 controlled paradigms. By training classifiers on hidden states from each layer of LLMs, we determine where syntactic competence emerges and stabilizes. For meaning and reasoning probing, we use the Conceptual Minimal Pair Sentences (COMPS) dataset [33], which tests conceptual understanding and property inheritance. Its controlled sentence-pair design allows us to distinguish between surface-level semantic association and inference-based reasoning. We further extend reasoning evaluation with ProntoQA [35] and WinoGrande [22] to assess multi-hop deductive and commonsense reasoning abilities respectively. These datasets together enable a hierarchical dissection of linguistic representations. Detailed dataset structures, task formulations, and examples are provided in Appendix A.

### 2.2 Language Model

Our main experiments use the Qwen2.5-14B model, a 14.7B-parameter transformer with 48 layers [36]. Each layer outputs hidden states of dimension 5120. The model supports a maximum context length of 131k tokens, though our experiments only use context sizes of 50 tokens. We use the base model without any instruction tuning or task-specific fine-tuning, and employ it both to probe the layer-wise emergence of linguistic and reasoning features and to perform brain encoding analyses.

To assess the robustness of our probing pipeline, we additionally examined other models across the Qwen family, spanning sizes from 1.8B to 14B parameters and multiple generations (Qwen1, Qwen1.5, Qwen2, Qwen2.5, and Qwen3) [4, 42, 36, 37]. We evaluated all models and found that Qwen2.5-14B exhibited the strongest reasoning capability, which is why we selected it as our primary model for subsequent analyses (see Appendix C.2).

### 2.3 Minimal Pair Probing

We apply minimal pair probing [14, 15] to identify the LLM layers most specialized for syntactic, meaning, and reasoning features. For each probing dataset, we extract sentence representations from each layer by feeding the sentence in isolation and taking the hidden state of its final token. We then use these sentence representations to train a logistic regression classifier to predict the correct item in each minimal pair. Model performance at a given layer is measured by the normalized accuracy score of this classifier.

Let $H_l \in \mathbb{R}^{n \times d}$ denote the matrix of sentence embeddings at layer $l$, where each row corresponds to the final-token hidden state of one sentence in the minimal pair dataset. To determine the most specialized layer for a given feature, we define the saturation layer as the earliest layer where performance plateaus:

$$L_x := \min\left\{l \mid \forall l' > l, \ Acc^{\mathcal{D}_x}(H_{l'}) - Acc^{\mathcal{D}_x}(H_l) < \varepsilon\right\}, \quad x \in \{s, m, r\}$$

where $\mathcal{D}_s = $ BLiMP, $\mathcal{D}_m = $ COMPS-BASE, and $\mathcal{D}_r = $ COMPS-WUGS-DIST, and $\varepsilon$ is a small threshold representing the tolerance for marginal improvement. The hidden states at the identified

---

**Algorithm 1** Construction of Feature-Specific Residual Embeddings

---

1: **Input:** LLM hidden states $\{H_L\}_{L=0}^{L_{\max}}$ for each token; probing datasets $\mathcal{D}_s, \mathcal{D}_m, \mathcal{D}_r$
2: **Output:** Feature-specific embeddings $E_l, E_s, E_m, E_r$
3: Perform probing with $\mathcal{D}_s, \mathcal{D}_m, \mathcal{D}_r$ to find saturation layers:
4:      $L_s \leftarrow$ syntax saturation layer from $\mathcal{D}_s$
5:      $L_m \leftarrow$ meaning saturation layer from $\mathcal{D}_m$
6:      $L_r \leftarrow$ reasoning saturation layer from $\mathcal{D}_r$
7: $L_l \leftarrow 0$
8: Define lexical embedding: $E_l \leftarrow H_{L_l}$»
9: **for** each $(L_{\text{low}}, L_{\text{high}})$ in $\{(L_l, L_s), (L_s, L_m), (L_m, L_r)\}$ **do**
10:      Train ridge regression $g$ to predict $H_{L_{\text{high}}}$ from $H_{L_{\text{low}}}$
11:      Compute residual embedding: $E \leftarrow H_{L_{\text{high}}} - g(H_{L_{\text{low}}})$
12:      Assign $E_s, E_m, E_r$ accordingly

---

saturation layers, denoted as $H_s := H_{L_s}$, $H_m := H_{L_m}$, and $H_r := H_{L_r}$, are used to construct feature-specific embeddings in the next stage.

## 2.4 Feature-Specific Embeddings

Building on the saturation layers identified via minimal pair probing, we construct four feature-specific embeddings to isolate distinct types of linguistic information: lexical, syntactic, meaning, and reasoning. Since hidden states at various layers contain overlapping linguistic information, we remove contributions from earlier representations to disentangle the targeted feature.

**Lexical embedding.** We define the lexical embedding $E_l \in \mathbb{R}^d$ as the hidden state at layer 0 of the LLM. As this layer follows token embeddings directly, it reflects uncontextualized lexical properties.

**Residual embeddings for syntax, meaning, and reasoning.** For the remaining features, syntactic, meaning, and reasoning, we construct residual embeddings by removing lower-level contributions from higher-layer representations. For instance, to isolate reasoning information, we remove the meaning contribution from the layer where reasoning saturates:

$$E_r := H_r - g_r(H_m), \quad \text{where } g_r = \arg\min_W \|H_r - W H_m\|_F^2 + \alpha \|W\|_F^2,$$

where $g_r$ is a ridge regression trained via 4-fold cross validation on a podcast corpus described in the next paragraph (2.4). The same procedure applies to compute $E_s$ and $E_m$. Specifically:

$$E_m = H_m - g_s(H_s) \qquad \text{(meaning minus syntactic)}$$
$$E_s = H_s - g_l(H_l) \qquad \text{(syntax minus lexical)}$$

This yields feature-specific representations that are aligned with the linguistic hierarchy (Appendix B) and minimally confounded by lower-level signals.

**Dataset for residual regression training.** To extract feature-specific residual embeddings, we train ridge regression models that require a large number of training samples. However, the transcript that will later be used for neural alignment, which is drawn from a single 30-minute podcast episode, is too short to support stable regression. To address this, we train the models on an expanded corpus of 16 transcribed episodes from the same podcast series, including the episode used in the alignment analysis [30]. This 160k-token dataset enables regression without PCA, preserving richer structure in the hidden states. We then apply the trained models to the target transcript to extract residuals for encoding analysis.

## 2.5 Encoding Model

**ECoG dataset.** After constructing feature-specific embeddings, we assess their neural alignment using the Podcast ECoG dataset [43]. This dataset contains high-gamma band (70–200 Hz) intracranial recordings from nine participants as they listened to a 30-minute narrative podcast. It includes 1,330

electrodes and a time-aligned word-level transcript, making it ideal for testing how lexical, syntactic, meaning, and reasoning embeddings align with neural activity during language comprehension.

As described in Section 2.4, we train ridge regression models on an expanded podcast corpus to isolate feature-specific residuals. We then apply these models to the ECoG-aligned transcript to extract feature-specific embeddings, which are used to predict neural responses. Each embedding is aligned to individual word onsets, and for each word, neural signals are epoched in a $\pm 2$s window and downsampled to 32 Hz, yielding $t = 128$ time bins per event. Let $X \in \mathbb{R}^{n \times d}$ be the matrix of input embeddings (lexical, syntactic, meaning, or reasoning) across $n$ word-aligned tokens, and $Y \in \mathbb{R}^{n \times (c \cdot t)}$ the corresponding ECoG response matrix across $c$ electrodes and $t$ time lags. We fit:

$$W^* = \arg\min_W \|Y - XW\|_F^2 + \alpha \|W\|_F^2,$$

with $\alpha$ selected via 5-fold cross-validation over a log-spaced grid, and $b = 5$ bootstrap resamples per fold using contiguous chunks of length $l = 32$. Model performance is quantified by Pearson correlation between predicted and actual signals. Temporal profiles are obtained by averaging over channels at each lag; spatial maps visualize per-channel peak correlations on 3D brain coordinates.

**Word-rate feature regress out.** We controlled for generic acoustic onset responses by adding a two-column word-rate covariate (word onsets and syllable-rate) to every ridge model. All variance-partitioning steps therefore quantify the variance explained beyond that attributable to mere word onsets. Let $R_{\text{full}}$ denote the cross-validated prediction correlation achieved using the combined feature set (embedding + word rate), and let $R_{\text{wc}}$ denote the correlation using only the word rate features. Assuming approximate orthogonality between the two feature sets, we estimated the unique contribution of the embedding features as: $R_{\text{embed}} = \text{sign}(R_{\text{full}}) \cdot \sqrt{\max(0, R_{\text{full}}^2 - R_{\text{wr}}^2)}$. This operation projects the full correlation vector onto the embedding-only axis, removing variance explained by word rate features. The assumption of orthogonality is approximately satisfied due to the preprocessing pipeline, and helps to prevent over-attribution of shared variance.

**Null distribution and responsiveness criterion.** Considering that different channels and features have varying signal-to-noise ratios (SNRs), we constructed a subject–electrode–specific null distribution to assess whether a feature block explains neural activity beyond chance and to enable cross-electrode analysis. This was done by shuffling the feature rows 500 times while keeping the word-onset covariates fixed. For every shuffle we refit the ridge model and recorded the peak correlation $R$. Because Pearson correlations are bounded and skewed near $\pm 1$, we applied the standard Fisher $z$-transform (`atanh`) to all correlations, computed the shuffle mean and s.d. in $z$-space, and converted the true correlation to a $z$-score. Electrodes with $z > 3.95$ (one-tailed $\alpha = .05$, Bonferroni-corrected across $N = 1268$ electrodes) were deemed responsive, corresponding to values exceeding 3.95 standard deviations above the shuffle mean.

## 3 Probing Results

Applying our probing pipeline to Qwen2.5-14B, we observe a clear progression in the emergence of representational features. Syntax saturates earliest at layer 6, followed by meaning at layer 20, and reasoning only at the deeper layer 30. This ordering highlights that low-level linguistic structure is captured quickly, whereas high-level reasoning requires substantially more depth.

To verify that this pattern is not specific to a single model, we extended the probing pipeline across multiple Qwen families spanning different sizes and generations. The same emergence order was consistently observed, with only one exception in Qwen-1.8B, where syntax and meaning saturated at the same layer. Full cross-model analyses are reported in Appendix B.

## 4 Disentanglement Validation

### 4.1 Matrix-Level Orthogonality of Residual Embeddings

We justify the approximate orthogonality of the feature-specific embeddings $E_l, E_s, E_m, E_r$ at the level of their embedding matrices. Each residual embedding can be represented as an $n \times d$ matrix, where $n$ is the number of tokens in the dataset and $d$ is the embedding dimension. Orthogonality here

is considered along the sample axis: the $n$-dimensional feature columns of one residual embedding matrix are approximately uncorrelated with those of another.

This follows from the progressive emergence of linguistic features across LLM layers (Table 1): syntax peaks early, meaning in mid layers, and reasoning in later layers. Once a feature reaches saturation, its accuracy score remains stable in deeper layers, indicating that later representations retain earlier features. As a result, representations like $H_m$ already embed information from $H_l$ and $H_s$, making regression from $H_m$ alone comparably as informative as from all three:

$$g_r(H_m) \approx g_r'([H_l, H_s, H_m]) \quad \Rightarrow \quad E_r \approx H_r - g_r'([H_l, H_s, H_m]) =: E_r'$$

Since $E_r'$ is the residual of a linear projection onto $[H_l, H_s, H_m]$, it is orthogonal to each:

$$E_r' \perp H_l, \quad E_r' \perp H_s, \quad E_r' \perp H_m$$

Each residual embedding $E_j \in \{E_l, E_s, E_m\}$ is a linear combination of earlier hidden states (e.g., $E_m = H_m - g_m(H_s) = H_m - W_m H_s$). By the bilinearity of covariance, we have:

$$\text{Cov}(E_j, E_r') = \text{Cov}(H_j - W_j H_k, E_r') = \text{Cov}(H_j, E_r') - W_j \text{Cov}(H_k, E_r') = 0$$

whenever $E_r' \perp H_j$ and $H_k$, for appropriate $j, k \in \{l, s, m\}$. This implies:

$$\langle E_r', E_j \rangle = 0 \quad \forall j \in \{l, s, m\}$$

Applying this proof across all residual stages, we conclude approximate mutual orthogonality of residual embedding matrices:

$$\langle E_j, E_k \rangle \approx 0 \quad \text{for all } j \neq k.$$

This sample-axis orthogonality ensures that residual embedding matrices contribute non-overlapping signals across the dataset, which is precisely the property required for the subsequent brain encoding analysis.

## 4.2 Token-Level Orthogonality via Cosine Similarity

While Section 4.1 establishes matrix-level orthogonality across the dataset, we also test orthogonality at the level of individual tokens. For each token $i \in \{1, \ldots, N\}$, let $E_l^i, E_s^i, E_m^i, E_r^i$ denote the four feature-specific residual vectors. We compute a $4 \times 4$ cosine similarity matrix $C_i$, take its absolute value $|C_i|$, and average across samples:

$$[C_i]_{j,k} = \frac{\langle E_j^i, E_k^i \rangle}{\|E_j^i\| \cdot \|E_k^i\|}, \quad \bar{C} = \frac{1}{N} \sum_{i=1}^N |C_i|, \quad j, k \in \{l, s, m, r\}$$

In the ideal case of perfect disentanglement, off-diagonal entries of $\bar{C}$ would be zero, indicating orthogonality between different embeddings. Figure 2a compares the pairwise mean absolute cosine similarity among raw hidden states at the four saturation layers (top) and among the corresponding residual embeddings (bottom). While the hidden states show substantial overlap, especially between meaning and reasoning layers ($\bar{C} = 0.751$), the residual embeddings exhibit near-zero off-diagonal similarity(all $<= 0.045$) across all pairs.

Thus, Section 4.1 and Section 4.2 together provide complementary evidence: residuals are linearly novel across the sample axis, and they also align with near-orthogonal directions at the token level. These findings confirm that residualization yields disentangled representations of lexical, syntactic, semantic, and reasoning information.

## 4.3 Feature Probing on Residual Embeddings

While mutual decorrelation ensures that residual embeddings are separated from one another, each residual should also preserve information relevant to its intended linguistic feature. We evaluate this by reapplying the same probing tasks used to define feature emergence. Since the lexical embedding $E_l$ is directly extracted rather than residualized, we focus on the syntactic ($E_s$), meaning ($E_m$), and reasoning ($E_r$) residuals.

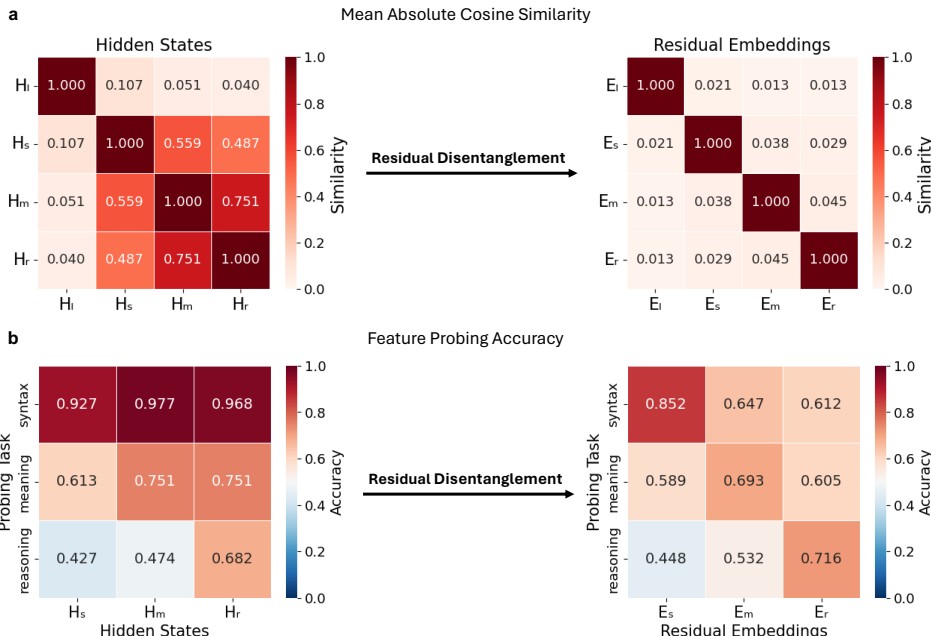

Figure 2: **a)** Pairwise cosine similarity among representations before (left) and after (right) residual disentanglement. The hidden states at feature saturation layers ($H_l, H_s, H_m, H_r$) exhibit substantial overlap. In contrast, the residual embeddings ($E_l, E_s, E_m, E_r$) show near-zero off-diagonal similarity. **b)** Feature probing results on hidden states (left) and residual embeddings (right), shown as accuracy. Each residual embedding achieves the highest performance on its corresponding task, while performing worse on unrelated tasks.

For each case, we train a logistic regression classifier on the corresponding residual embedding and report accuracy scores:

$$\hat{y}_i = \sigma(w_j^\top E_j^i + b_j), \quad j \in \{s, m, r\},$$

where $\sigma(\cdot)$ is the sigmoid function and $E_j^i \in \mathbb{R}^d$ is the residual embedding for token $i$.

Results are shown in Figure 2b. Each residual achieves the highest performance on its own task (e.g., $E_s$ on syntax probing, $E_m$ on meaning, $E_r$ on reasoning) while performing worse on unrelated tasks. The diagonal entries retain accuracy similar to the hidden-state baseline, whereas the off-diagonal entries drop substantially, with decreases of 33.8% (syntax task evaluated on the meaning residual), 36.8% (syntax task evaluated on the reasoning residual), and 19.4% (meaning task evaluated on the reasoning residual). These findings demonstrate that the residual embeddings are meaningfully disentangled and capture feature-specific information rather than reflecting general task complexity. A more detailed analysis of cross-task accuracies, including Bag-of-Word baseline, is provided in Appendix D.

**Summary of Disentanglement Validation** Across three complementary tests (matrix-level orthogonality proof, token-wise orthogonality computation, and probing on the residual embeddings), residualization consistently produces four nearly orthogonal, interpretable embeddings corresponding to distinct linguistic levels. These embeddings form the basis for our subsequent neural encoding experiments.

## 5 Encoding Results

### 5.1 Shallow features dominate neural prediction, but lexical activations are sparse.

Across time-aligned word events, encoding models based on shallower linguistic features, such as lexical identity and syntactic structure, showed the strongest neural correlations (Figure 3). Electrodes

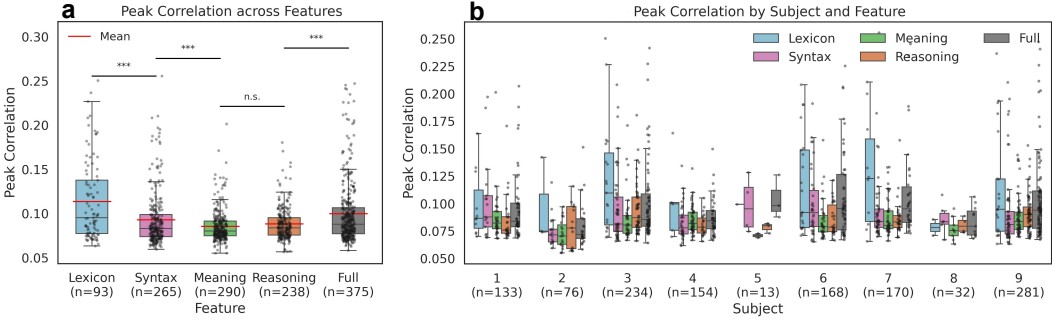

Figure 3: **a)** Peak correlation across linguistic feature models. As described in the Methods section, electrodes with z > 3.95 (one-tailed $\alpha = .05$, Bonferroni-corrected across N = 1268 electrodes) were deemed responsive, corresponding to values exceeding 3.95 standard deviations above the shuffle mean. Boxplots indicate the median and interquartile range across activated electrodes with red lines indicating means additionally. Asterisks mark significant differences between adjacent features (Welch's t-test; $p < 0.001$), showing that lexical and syntax features yield significantly higher correlations than high-level linguistic representations. **b)** Peak correlation by subject and feature. The number of responsive electrodes for each subject is shown below the x-axis. Lexical features generally exhibit higher correlations, indicating stronger neural alignment for lower-level linguistic representations.

responsive to these features exhibited markedly higher peak correlation values than those driven by meaning or reasoning embeddings (Welch's t-test, $p < 0.001$), confirming that lower-level linguistic information accounts for a disproportionate share of brain–model alignment under linear encoding. However, despite its high correlation strength, the lexicon embedding activated the fewest electrodes overall, suggesting that its predictive precision arises from a smaller, more specialized subset of cortical sites. In contrast, deeper features such as meaning and reasoning recruited broader but less strongly tuned populations. This dissociation between activation magnitude and spatial extent indicates that shallow representations, though limited in coverage, achieve tighter alignment within localized neural circuits, whereas deeper representations engage distributed cortical networks with weaker linear correspondence. Moreover, without disentanglement, the temporal dynamics in Figure 4 reveal that the full embedding is dominated by lexical and syntactic signals, whose early and sharp peaks mask the slower, high-level temporal signatures of meaning and reasoning. This highlights the necessity of residual separation for isolating deeper cognitive processes from low-level linguistic confounds.

## 5.2 Reasoning embedding shows different temporal pattern compared to Lexicon, Syntax and Meaning.

The temporal correlation profiles of Figure 4 illustrate a clear processing hierarchy among linguistic representations. Syntactic signals exhibit an early rise that begins before word onset and reach their maximum slightly pre-onset, suggesting that a large portion of predictive activity for upcoming words might be syntactically driven. Lexical features show a sharp and temporally confined peak immediately after onset, consistent with rapid low-level mapping between acoustic input and lexical identity. Meaning representations start to increase even before onset but peak afterward, indicating that contextual semantics contribute both to anticipatory prediction and to continued integration once the word is heard. In contrast, reasoning signals begin to rise only after onset and reach their peak at around 362 ms, reflecting a delayed, high-level computation that likely involves integrative or inferential processes beyond immediate linguistic parsing. Finally, the full embedding closely tracks the envelope of the four disentangled features, mirroring their composite dynamics and reinforcing the reliability of our residual separation procedure. Together, these temporal patterns reveal a cascading progression from predictive syntactic structure to contextual semantics, and finally to post-lexical reasoning operations in the human cortex.

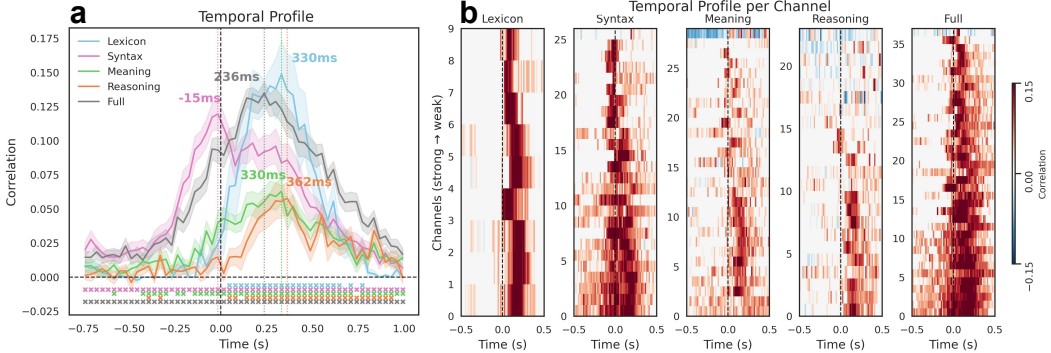

Figure 4: **a)** Average correlation time courses for the top 10% most responsive electrodes reveal distinct temporal dynamics across linguistic features. Responsive electrodes were identified following the strategy described in the Methods section. Syntactic signals rise before word onset, followed by meaning, lexicon and reasoning, which peak latest at ∼362 ms. Significance markers (×) indicate time points where correlations are significantly greater than zero (one-tailed $t$-test, FDR-corrected $q < 0.05$). **b)** Temporal profile of individual electrodes selective to each feature.: Lexicon, Syntax, Meaning, Reasoning, and Full. Electrodes were selected the same way as in panel a.

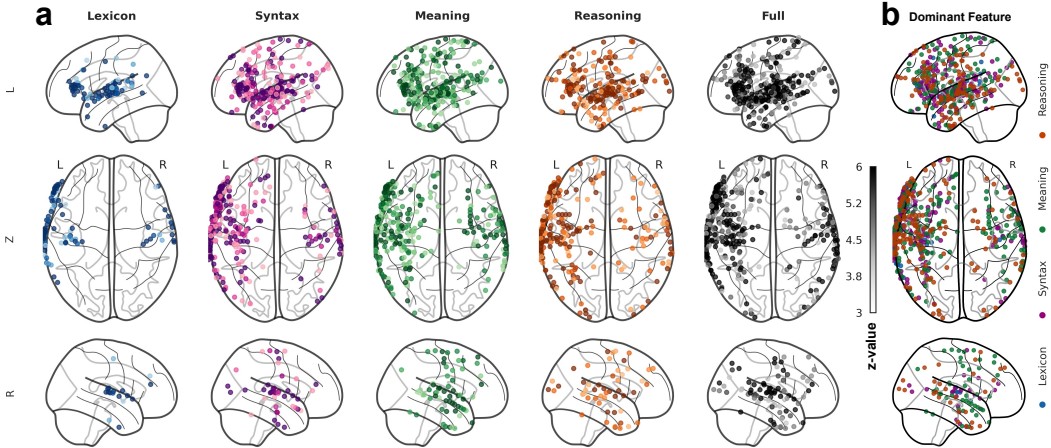

Figure 5: **a)** Spatial distribution of feature-selective responsive electrodes (MNI space). Displayed electrodes are those that met the responsiveness criterion defined in the Methods section. For visualization, z-scores were clipped to 3–6 to prevent extreme values from dominating the color scale and obscuring spatial patterns. **b)** For each electrode, the feature with the highest peak z-score among Lexicon, Syntax, Meaning, and Reasoning models is shown as its dominant feature. Colored dots indicate the dominant model. Across all electrodes, the counts of dominant features were: Syntax = 166, Meaning = 161, Reasoning = 128, and Lexicon = 42.

## 5.3 Reasoning recruits more than language area compared to low-level aspects.

As shown in Figure 5, shallow linguistic features such as lexicon exhibit high peak correlations but remain confined to classical language regions, including the inferior frontal gyrus (IFG) and superior temporal gyrus (STG). The strong yet spatially restricted activations indicate that lexical representations are predominantly localized within the traditional perisylvian language network, reflecting low-level linguistic encoding. In contrast, syntax, meaning, and particularly reasoning embeddings engage a broader set of cortical regions. While all three maintain robust responses in IFG and STG, reasoning-selective electrodes extend anteriorly into the superior frontal gyrus (SFG) and posteriorly into the visual cortex, showing significant activation compared with other

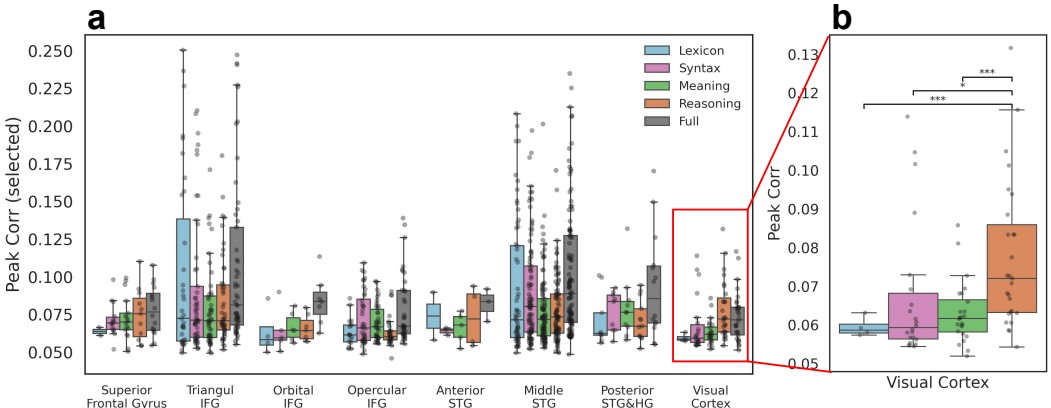

Figure 6: **a)** Peak correlation across cortical regions and feature models. Responsive electrodes were selected based on bilateral significance above the null shuffle distribution, $\alpha < 0.05$. The analysis focuses on language-related cortical regions, including the inferior frontal gyrus (IFG), superior temporal gyrus (STG), and surrounding Heschl's Gyrus (HG), as well as the superior frontal gyrus and visual cortex. The visual cortex group includes electrodes located in the superior occipital sulcus, transverse occipital sulcus, calcarine sulcus, occipital pole, superior occipital gyrus, middle occipital gyrus, and inferior occipital gyrus and sulcus. **b)** Welch's t-test (one-tailed) revealed that electrodes in visual cortex activated by the Reasoning feature showed significantly higher correlations than those activated by all other features ($p < 0.001$ for '***', $p < 0.05$ for '*').

features (Welch's t-test; p < 0.001; Figure 6). This expanded recruitment beyond canonical language areas suggests that reasoning might involve higher-order cognitive operations that integrate linguistic input with abstract inference, visual imagery, and executive control. Such cross-modal engagement underscores that reasoning-related brain activity cannot be fully explained by linguistic processing alone, but rather reflects the neural substrates of generalizable, multimodal cognition[25, 12].

# 6   Discussion

By isolating distinct types of linguistic information using residual embeddings, we disentangled the contributions of lexicon, syntax, meaning, and reasoning to the neural encoding of natural speech. This revealed new insights into the spatial and temporal organization of linguistic features that cannot be captured by full LLM embeddings used in prior studies [20, 38, 13, 34, 17, 7, 27, 2, 3, 8, 9, 32, 6]. Previous work has tried hierarchical approaches to decompose low-level linguistic and acoustic features in the brain (Keshishian et al. [23, 24]). However, our study is the first to apply hierarchical disentanglement framework to large language models by first localizing feature emergence layers through probing and then performing residual decoupling to construct layer-wise feature representations, thereby demonstrating its utility for isolating higher-order reasoning representations.

Our findings show that lower-level features, such as syntax and lexicon, elicit stronger neural activations than higher-level features like meaning or reasoning. This reflects both the stronger linear accessibility of shallow features and an inherent bias in full LLM embeddings, which primarily capture syntactic regularities while obscuring deeper computations [26]. Residual disentanglement reveals clear spatiotemporal hierarchies: shallow features emerge earlier and localize to classical language regions (IFG, STG), whereas reasoning signals arise later and extend beyond the perisylvian network into SFG and visual cortex. These results indicate that reasoning engages broader cortical systems that integrate linguistic processing with higher-level cognitive and multimodal functions. Together, our framework provides a principled approach for separating linguistic and cognitive components in LLMs and mapping them onto the human brain, offering an alignment-based perspective on how language and reasoning may correspond between artificial and neural systems.

## Acknowledgment

This work is funded by the National Institutes of Health (NIH- NIDCD) and a grant from Marie-Josée and Henry R. Kravis.

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

# A  Detailed Description of Probing Datasets

**BLiMP for syntactic probing.**  To construct feature-specific representations for brain alignment, we first identify LLM layers that specialize in syntactic encoding. Minimal pairs and probing techniques have been widely used in NLP field to access LLM's internal representation [31, 28, 11, 18, 29, 16]. Here we use the Benchmark of Linguistic Minimal Pairs (BLiMP) [40], which evaluates syntactic knowledge through controlled minimal pairs differing in grammaticality across 67 paradigms. Each pair isolates a specific grammatical phenomenon such as subject–verb agreement, reflexive anaphora, or negative polarity. By training a classifier to distinguish acceptable from unacceptable sentences using hidden states from each LLM layer, we detect where syntactic competence emerges and peaks. For example, in a task targeting subject–verb agreement:

 a. The cats annoy Tim. ✓
 b. The cats annoys Tim. ✗

We average classifier performance across all paradigms to determine the saturation layer where syntactic features are stably encoded.

**COMPS for Meaning and Reasoning Probing.**  For meaning and reasoning probing, we use the Conceptual Minimal Pair Sentences (COMPS) dataset [33]. Sentence pairs in COMPS differ in whether the subject plausibly inherits a property. By probing whether each layer favors the correct concept-property association, we detect the layers where meaning and reasoning abilities peak.

COMPS-BASE evaluates surface-level understanding without requiring inference. For example, given the property "can heat food":

 a. An oven can heat food. ✓
 b. A refrigerator can heat food. ✗

COMPS-WUGS-DIST increases reasoning complexity by replacing known concepts with nonsense words (e.g., "wug", "dax") and inserting distractor sentences that introduce interference. This design prevents the model from relying on surface-level co-occurrence or positional cues: distractors can be reordered. The model must infer the identity of the nonsense word from context to generalize property inheritance.

 a. A wug is an oven. A dax is a refrigerator. Therefore, a wug can heat food. ✓
 b. A wug is an oven. A dax is a refrigerator. Therefore, a dax can heat food. ✗

This progressive design increases reasoning difficulty and allows us to localize where models transition from semantic to reasoning-level representation.

**Additional reasoning benchmarks.**  Beyond COMPS, we incorporate two additional reasoning benchmarks to validate robustness across task formats: ProntoQA [35], which evaluates multi-hop deductive reasoning, and WinoGrande [22], which targets commonsense coreference resolution. For ProntoQA, we only use examples that require 5-hop deductive inferences to ensure high difficulty, where solving the task requires maintaining consistent symbolic relations across multiple premises and performing compositional logical deduction. Two illustrative examples are:

 a. Dogs are cats. Each dog is sour. Vertebrates are dull. Felines are dogs. Felines are not dull. Cows are felines. Each cow is aggressive. Snakes are cows. Snakes are orange. Animals are snakes. Every animal is not luminous. Mammals are animals. Each mammal is hot. Fae is a mammal. Therefore, Fae is not dull. ✓
 b. Every carnivore is a dog. Carnivores are angry. Every mammal is a carnivore. Each mammal is not red. Each snake is a mammal. Each snake is transparent. Cats are snakes. Cats are nervous. Each sheep is not angry. Each animal is a cat. Animals are earthy. Sam is an animal. Therefore, Sam is not angry. ✗

For WinoGrande, which tests commonsense reasoning through contextual coreference resolution, the model must leverage world knowledge and pragmatic inference to identify the correct referent under minimal lexical cues. A pair of examples is:

a. John moved the couch from the garage to the backyard to create space. The garage is small. ✓
  b. John moved the couch from the garage to the backyard to create space. The backyard is small. ✗

Together, these datasets complement COMPS by spanning property inheritance, deduction, and commonsense inference. Detailed experiments and cross-validation results are provided in Appendix C.

**Bad Probing Task Filtering**   We employ a Bag-of-Words (BoW) baseline and exclude tasks where the BoW model achieves an accuracy above 0.6, resulting in a final set of 29 BLiMP tasks (out of 67). The meaning (COMPS-BASE) and reasoning (COMPS-WUGS-DIST) tasks remain unchanged.

This filtering step is motivated by the fact that BoW readily captures tasks solvable via simple lexical cues, indicating a shallow design. By removing these superficially "easy" tasks, we ensure that the retained tasks demand genuine syntactic or semantic understanding rather than mere lexicon-based heuristics.

## B   Hierarchical Emergence of Linguistic Features

In the main text, we reported probing results for Qwen2.5-14B (Section 3), showing that syntax, meaning, and reasoning features saturate at layers 6, 20, and 30, respectively. To evaluate the robustness of this emergence pattern, we extended our probing pipeline to 17 models across the Qwen family, including Qwen, Qwen1.5, Qwen2, Qwen2.5, and Qwen3 models.

Across nearly all models, we observed the same progression of feature emergence: syntax saturates earliest, followed by meaning, and then reasoning. The only exception was Qwen-1.8B, where syntax and meaning saturated at the same layer. Table 1 reports the saturation layers identified for all models.

Table 1: Saturation layers for syntax, meaning, and reasoning across Qwen models.

| Model | Syntax | Meaning | Reasoning |
|---|---|---|---|
| Qwen-1.8B | 11 | 11 | 14 |
| Qwen-7B | 11 | 13 | 15 |
| Qwen-14B | 9 | 16 | 20 |
| Qwen1.5-1.8B | 10 | 11 | 14 |
| Qwen1.5-7B | 9 | 13 | 16 |
| Qwen1.5-14B | 8 | 16 | 20 |
| Qwen2-1.5B | 10 | 14 | 17 |
| Qwen2-7B | 7 | 14 | 18 |
| Qwen2.5-1.5B | 7 | 14 | 18 |
| Qwen2.5-7B | 7 | 15 | 19 |
| Qwen2.5-14B | 6 | 20 | 30 |
| Qwen3-1.7B | 9 | 16 | 19 |
| Qwen3-8B | 7 | 20 | 23 |
| Qwen3-14B | 5 | 17 | 27 |

## C   Consistency Across Reasoning Benchmarks

In the main text, reasoning saturation layers were identified using COMPS-WUGS-DIST. To evaluate robustness across tasks of varying complexity, we additionally applied two further reasoning probes: a 5-hop deductive reasoning task from ProntoQA and the WinoGrande benchmark for commonsense coreference.

We tested all three probes, COMPS-WUGS-DIST, ProntoQA, and WinoGrande, across 17 Qwen models spanning multiple sizes and training modes. Results show strong consistency: the average difference in reasoning saturation layers between COMPS-WUGS-DIST and ProntoQA was $0.94$, and between COMPS-WUGS-DIST and WinoGrande was $0.53$. Given overall model depths of 25–49 layers, these differences are negligible and support the robustness of our reasoning probe.

## C.1 Consistent Saturation Layers Across Reasoning Benchmarks.

Table 2 reports the reasoning saturation layers identified by each probe. The high degree of agreement across the three tasks indicates that reasoning-specific representations are consistently localized in deep layers, regardless of task format. The agreement across COMPS-WUGS-DIST, ProntoQA, and WinoGrande confirms that our identification of reasoning-specific saturation layers generalizes beyond a single benchmark. This consistency reinforces the conclusion that reasoning features emerge reliably in deep layers across architectures and reasoning formats.

Table 2: Reasoning saturation layers identified by different probes across Qwen models.

| Model | ProntoQA | COMPS-WUGS-DIST | WinoGrande |
|---|---|---|---|
| Qwen-1.8B | 13 | 14 | 14 |
| Qwen-7B | 13 | 15 | 16 |
| Qwen-14B | 18 | 20 | 20 |
| Qwen1.5-1.8B | 13 | 14 | 14 |
| Qwen1.5-7B | 14 | 16 | 16 |
| Qwen1.5-14B | 20 | 20 | 22 |
| Qwen2-1.5B | 17 | 17 | 17 |
| Qwen2-7B | 16 | 18 | 19 |
| Qwen2.5-1.5B | 16 | 18 | 17 |
| Qwen2.5-7B | 17 | 19 | 19 |
| Qwen2.5-14B | 29 | 30 | 29 |
| Qwen3-1.7B | 19 | 19 | 19 |
| Qwen3-8B | 23 | 23 | 23 |
| Qwen3-14B | 27 | 27 | 26 |

## C.2 Qwen2.5-14B Achieves the Best Reasoning Performance (Small-Scale Models)

We evaluated the Qwen family on three reasoning benchmarks: ProntoQA (multi-hop deductive reasoning), COMPS-WUGS-DIST (property inheritance with distractors), and WinoGrande (commonsense coreference). Table 3 reports accuracy on each benchmark, along with the average across tasks. We observe consistent improvements in reasoning performance with increasing model scale and newer Qwen generations. Among the base models, Qwen2.5-14B achieves the highest average accuracy across benchmarks, reflecting its stronger reasoning capability, which is why we select it as the primary model for our experiments. These results complement our saturation layer analysis by showing that models allocating greater depth to reasoning layers also demonstrate superior task-level reasoning performance.

Table 3: Reasoning task performance across Qwen models. The Average column reports the mean accuracy across ProntoQA, COMPS-WUGS-DIST, and WinoGrande.

| Model | ProntoQA | COMPS-WUGS-DIST | WinoGrande | Average |
|---|---|---|---|---|
| Qwen-1.8B | 0.797 | 0.522 | 0.523 | 0.614 |
| Qwen-7B | 0.886 | 0.641 | 0.602 | 0.710 |
| Qwen-14B | 0.880 | 0.695 | 0.647 | 0.741 |
| Qwen1.5-1.8B | 0.792 | 0.513 | 0.523 | 0.609 |
| Qwen1.5-7B | 0.848 | 0.667 | 0.603 | 0.706 |
| Qwen1.5-14B | 0.910 | 0.670 | 0.653 | 0.744 |
| Qwen2-1.5B | 0.784 | 0.586 | 0.552 | 0.640 |
| Qwen2-7B | 0.851 | 0.636 | 0.660 | 0.716 |
| Qwen2.5-1.5B | 0.783 | 0.605 | 0.566 | 0.651 |
| Qwen2.5-7B | 0.879 | 0.673 | 0.664 | 0.739 |
| Qwen2.5-14B | **0.922** | 0.691 | **0.698** | **0.770** |
| Qwen3-1.7B | 0.739 | 0.500 | 0.530 | 0.589 |
| Qwen3-8B | 0.876 | 0.629 | 0.651 | 0.719 |
| Qwen3-14B | 0.912 | **0.716** | 0.670 | 0.766 |

# D  Clarification on Residual Embedding Probing Results

In Section 4.3, we reported that each residual embedding performs best on its corresponding task while performing worse on unrelated tasks. Some cross-task accuracies remain non-trivial, particularly on the meaning and reasoning probes. We provide additional clarification here.

**Lexical-level baseline.**  A control experiment with a bag-of-words (BoW) model, which ignores syntax and word order, achieved 0.665 accuracy on COMPS-BASE. This shows that lexical cues alone provide a strong baseline for the meaning task, well above random chance (0.5). In this context, the raw scores of the syntax and reasoning residuals on COMPS-BASE (0.589 and 0.605, respectively) are in fact *below* the BoW baseline, indicating that residualization successfully removes spurious lexical and structural information from the meaning embedding. This control confirms that the observed cross-task scores are largely explained by lexical baselines artifacts. The syntax, meaning, and reasoning residuals each retain information most relevant to their target feature, supporting the interpretation that residual embeddings are meaningfully disentangled.

# E  Supplemented Neuroscience Analysis

## E.1  Activated-electrode Overlap Across Linguistic Feature Models

Figure 7. Activated-electrode overlap across feature models. Each cell shows the number of electrodes that were identified as active in both the feature model indicated by the row and that indicated by the column. Diagonal values represent the total number of electrodes active within each model, while off-diagonal values quantify cross-feature overlaps. Notably, high overlaps between Syntax and Meaning (143) and between Full and all subcomponents (around 180) suggest that electrodes responsive to linguistic processing are largely shared across multiple feature spaces. This pattern indicates that while distinct feature models capture partially unique neural activations, substantial commonality exists among them, particularly for integrative language representations.

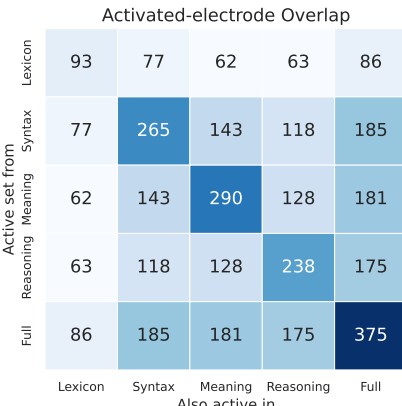

Figure 7: **Activated-electrode overlap across feature models.** Each cell shows the number of electrodes active in both feature sets, revealing substantial overlap among higher-level linguistic representations.

## E.2  Lateralization Analysis

To quantify hemispheric lateralization, we computed the proportion of active electrodes relative to total electrodes in the left and right hemispheres for each linguistic feature. Although the total number of electrodes is highly asymmetric across hemispheres (1057 on the left vs. 210 on the right), the normalized activation proportions still reveal clear trends. Specifically,

Table 4: **Hemispheric lateralization ratios.** Each proportion represents the fraction of active electrodes relative to total electrodes within each hemisphere (1057 left vs. 211 right). The ratio indicates $\frac{\text{Left proportion}}{\text{Right proportion}}$.

| Feature | #L | #R | Prop.L | Prop.R | L/R |
|---|---|---|---|---|---|
| Lexicon | 76 | 17 | 0.07 | 0.08 | 0.89 |
| Syntax | 225 | 41 | 0.21 | 0.20 | 1.10 |
| Meaning | 227 | 63 | 0.21 | 0.30 | 0.72 |
| Reasoning | 193 | 51 | 0.18 | 0.24 | 0.76 |
| Full | 300 | 79 | 0.28 | 0.37 | 0.76 |

Syntax-related encoding shows a higher proportion of
left-hemisphere activations (0.21 vs. 0.20), resulting
in the highest left/right ratio (1.09) among all fea-
tures. In contrast, Meaning and Reasoning features
exhibit relatively stronger right-hemisphere activations (ratios $< 1$). Given the uneven electrode
coverage across hemispheres, these findings should not be overinterpreted as true right-lateralization
for semantic or reasoning processing; however, they do suggest a stronger left-lateralization effect for
shallow representations (lexicon and syntax).

### E.3 Activated Channels in All Brain Area

Please refer to Figure 8.

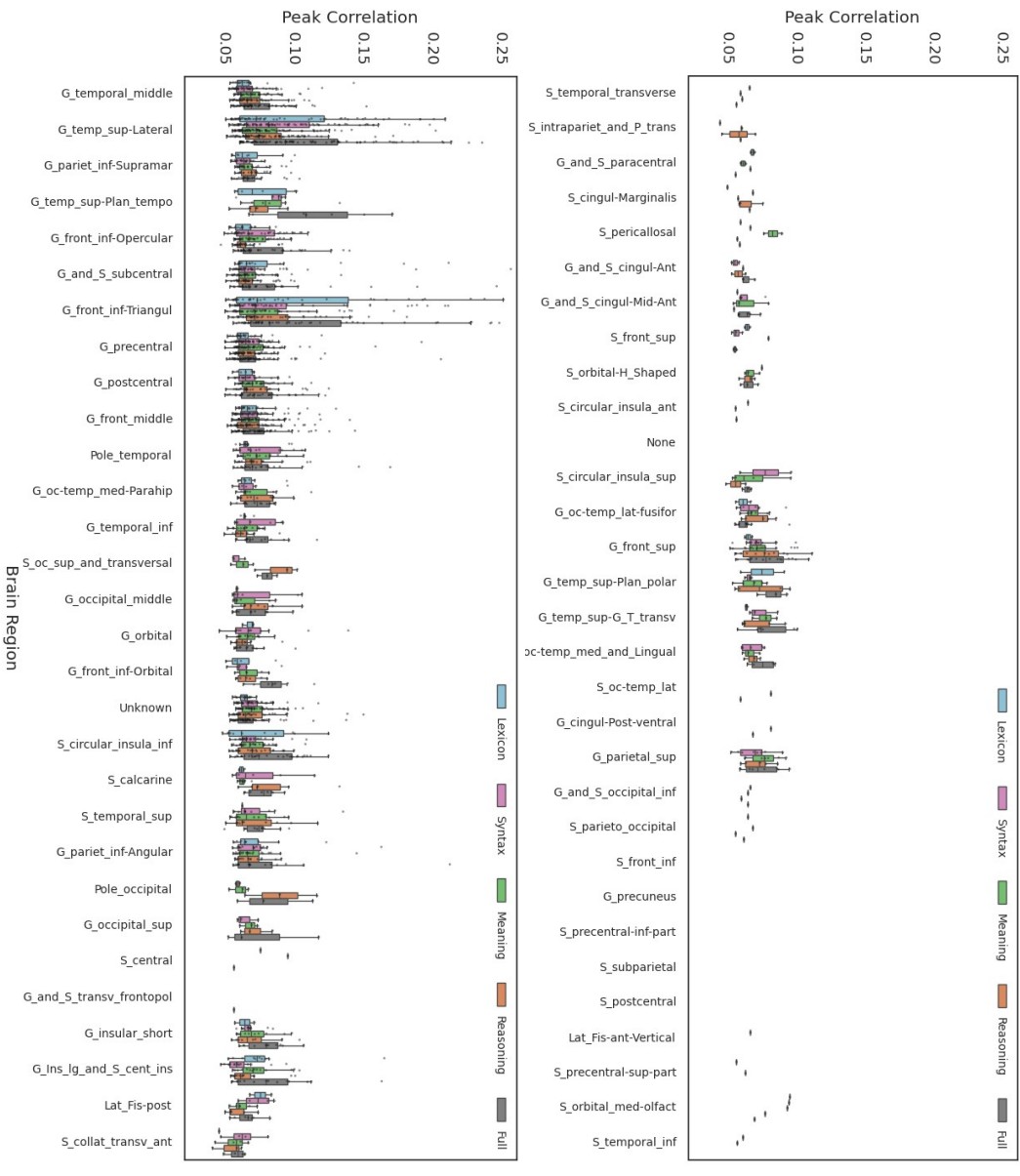

Figure 8: Peak correlation values of encoding performance across all cortical regions for five feature
types. Responsive electrodes were selected based on bilateral significance above the null shuffle
distribution, $\alpha < 0.05$.

## F  Limitations and Future Directions

Despite these advances, several limitations remain. Our reliance on ECoG data, although temporally precise, provides limited spatial coverage, with insufficient sampling in frontal regions that are crucial for reasoning. Future work should integrate complementary modalities such as fMRI to improve spatial resolution. Also, while Qwen2.5-14B balances scale and accessibility, larger models optimized for multi-step reasoning may reveal stronger or more specialized reasoning signals. While reasoning representations saturate around layer 30 in Qwen2.5-14B, the role of subsequent layers remains unclear. We hypothesize that deeper layers may encode more abstract or generative representations, potentially related to discourse organization, narrative planning, or world modeling, beyond explicit reasoning. Exploring how these higher-level representations might align with cortical hierarchies or large-scale cognitive organization in the brain presents an especially interesting direction for future work.

## G  Compute Resources and Execution Time

We report here the compute infrastructure and approximate runtime for each stage of our pipeline to support reproducibility.

- **Hidden State Extraction**: Performed using $4\times$ NVIDIA L40 GPUs. Extracting hidden states for around 164,000 tokens from the Qwen2.5-14B model. The extraction took approximately 40 minutes and required around $4 \times 30$ GB of GPU memory.
- **Layer-wise Probing**: Conducted using $1\times$ NVIDIA L40 GPUs. Each BLiMP task paradigm took 2.2 minutes on average, totaling around 57.2 minutes for the selected 26 paradigms focusing on syntax. COMPS-BASE consisting of 49,340 English sentence pairs took 1 hour and 47 minutes on average. COMPS-WUGS-DIST consisting of 27,792 pairs took 1 hour on average.
- **Residual Embedding Construction**: Ridge regression training was done using Scikit-learn on CPU with 30 GB of memory. Each regression model took less than 10 minutes to converge. NVIDIA cuML and GPU training was not adopted due to lack of support for multi-output ridge regression training.
- **Brain Encoding (ECoG)**: Conducted using CPU with self-developed package. Aligning model activations and training encoding models over all channels and time lags took approximately 5 hours for all experiments done.

Overall, the full pipeline can be reproduced on a modern workstation or cloud instance equipped with 1 NVIDIA L40 GPU + 30 GB RAM.

## H  Licenses and Attribution of External Assets

This work uses several publicly available datasets and a language model, all of which are properly cited and used in accordance with their respective licenses. No proprietary or scraped data was used.

- **BLiMP Dataset** [40]: A suite of syntactic probing tasks for language models.
  - URL: `https://github.com/alexwarstadt/blimp`
  - License: CC-BY 4.0
- **COMPS Datasets** [33]: COMPS-BASE and COMPS-WUGS-DIST are used to assess semantic and reasoning representations.
  - URL: `https://github.com/kanishkamisra/comps/`
  - License: Apache License 2.0
- **The Podcast ECoG Dataset** [43]: High-resolution electrocorticographic recordings from participants listening to a natural podcast.
  - URL: `https://openneuro.org/datasets/ds005574/versions/1.0.2`
  - License: CC0

- **Expanded Podcast Transcripts** [30]: Text for additional podcast episodes used to extend ECoG analysis.
    - URL: `https://github.com/calclavia/tal-asrd`
    - License: No explicit license is specified on the repository. However, the author provides access to the dataset and indicates that it can be downloaded for research purposes. We used the dataset solely for non-commercial academic research and did not redistribute or modify it.
- **Qwen2.5-14B** [36]: Language model for LLM representations extraction.
    - URL: `https://huggingface.co/Qwen/Qwen2.5-14B`
    - License: Apache license 2.0

All datasets and model were used solely for research purposes. We did not modify or redistribute any datasets beyond standard preprocessing steps for experimental use. All license terms were respected.

