# OpenReview forum: "Far from the Shallow: Brain-Predictive Reasoning Embedding through Residual Disentanglement"
_NeurIPS.cc/2025/Conference — NeurIPS 2025 poster_

### Official Review · Reviewer_hFzy · 2025-06-23

**Clarity:** 3
**Significance:** 3
**Originality:** 3
**Rating:** 4
**Confidence:** 4

**Summary:**

This paper presents a residual disentanglement method that identifies feature saturation layers through probing tasks, and using ridge regression to remove lower-level contributions from higher-level features. Results demonstrate that a "reasoning" embedding predicts a unique component of brain activation in human ECoG recordings (with later timing and engagement of frontal regions) that is obscured when using conventional embeddings.

**Questions:**

1. I would suggest testing across the Qwen model family (from smaller 1.8B model to 72B parameters) to provide convergent validity for the reasoning residuals. You should predict that smaller models with poorer reasoning performance should show weaker or absent "reasoning" residuals in the disentanglement pipeline, while larger models with demonstrated reasoning capabilities should yield stronger, more distinct reasoning embeddings, and the brain encoding performance of reasoning residuals should scale with the source model's actual reasoning abilities? If a 1.8B model's "reasoning" residuals predict brain activity just as well as a 72B model's, it would suggest they're capturing more generic complexity rather than genuine reasoning processes. This would also help test whether the layer localization methodology (finding reasoning "saturation" layers) generalizes across model scales, and whether the temporal dynamics (reasoning signals peaking ~330ms) hold only for models that actually demonstrate reasoning capabilities.

2. The COMPS-WUGS-DIST task represents a somewhat impoverished view of "reasoning" on its own. Simple property inheritance ("a wug is an oven -> a wug can heat food") and variable binding captures only basic logical inference, not the multi-step, abstract, or causal "situation model" reasoning typically associated with higher-order cognition. The limits the interpretability of the "reasoning" residuals and their neural correlates. I'd suggest using a more diverse set of reasoning benchmarks (e.g., winogrande, multi-step arithmetic) to validate that what are being called "reasoning" residuals generalize across diverse reasoning tasks as opposed to being an artifact of this one task instantiation.

3. I'd appreciate a stronger justification for why various assumptions of the residual disentanglement approach should hold empirically. For example, (a) the assumption that the syntax-> meaning -> reasoning hierarchy emerges in a strict sequence doesn't necessarily corresond to how modern transformers process information, (b) the (implicit) assumption that higher-level features are linear combinations of lower-level ones plus orthogonal residuals, (c) the assumption that features "saturate" at specific layers and remain stable thereafter (what looks like saturation might just be the limits of the probing task), and (d) while low cosine similarity between residuals tests orthogonality, this doesn't validate that the residuals capture the intended cognitive constructs -- just because they're orthogonal doesn't mean they correspond to cognitively meaningful categories, they might just be isolating arbitrary statistical patterns that happen to correlate with these specific reasoning tasks.

4. I would consider directly testing whether the extracted "reasoning" residuals actually causally contribute to reasoning performance. For example, if you corrupt or ablate the reasoning residuals in the original model, does reasoning performance selectively decline while other linguistic capabilities remain intact? This would provide much stronger evidence that the residuals are partitioning out functionally relevant reasoning representations.

**Ethical Concerns:**

["NO or VERY MINOR ethics concerns only"]

**Final Justification:**

After consideration of the authors' rebuttal, I am updating my score from 3 (Borderline reject) to 4 (Borderline accept).

The demonstration of convergent saturation layers with ProntoQA substantially strengthens the claim that the method identifies reasoning representations across different task types (not just COMPS). Additionally, extending the probing analysis across 17 Qwen models (1.8B-14B) provides stronger evidence for the generalizability of the syntax -> meaning -> reasoning hierarchy to models of different depths.

The following issues I raised still remain unresolved: (1) even with the inclusion of ProntoQA, the reasoning tasks still represent relatively narrow logico-deductive reasoning (vs physical or commonsense reasoning). So claims about isolating "reasoning" more broadly would benefit from validation on more diverse benchmarks. (2) two suggested experiments remain absent -- testing whether brain encoding performance scales with model reasoning performance capabilities, and causal ablation studies to establish the functional relevance of the residuals.

**Limitations:**

Yes.

**Quality:**

3

**Strengths And Weaknesses:**

Strengths:

The technical execution is rigorous, employing solid ECoG data and a novel residual disentanglement pipeline that includes careful probing, validation of orthogonality, and encoding analyses. The use of a capable modern language model (Qwen-2.5-14B) as opposed to the smaller models used in a lot of other recent work enables the detection of emergent reasoning capabilities that would be impossible with smaller models.The results reveal intriguingly distinct spatiotemporal activation patterns that suggest meaningful differences between linguistic features.

Weaknesses:

I'm concerned that the paper's central claim about isolating "reasoning" representations is stronger than what the evidence supports. First, the operationalization of "reasoning" through a single, simple task probe (COMPS-WUGS-DIST) does not capture the kind of abstract, multi-step reasoning typically associated with higher-order cognition, limiting the interpretability of the purported reasoning residuals and the corresponding neural analyses. The exciting possibility that these residuals capture genuine reasoning processes requires substantially stronger evidence across diverse reasoning benchmarks and models (see below).

---

> ### Author Rebuttal · Authors · 2025-07-31
>
> ## Response to Reviewer 4 (hFzy)
> **Concern 1: Reasoning Probes Rely Only on COMPS**
>
> **Summary of Response:** To strengthen our evaluation, we supplemented COMPS-WUGS-DIST with a more complex 5-hop reasoning task from ProntoQA. Applied across 17 Qwen models, the two tasks yielded closely aligned reasoning saturation layers.
>
> **Full Response:** Thank you for raising this point. Our original analysis used COMPS-WUGS-DIST, a 1-hop reasoning task, to identify reasoning saturation layers. While effective at detecting the presence of reasoning, its simplicity may limit generalizability.
>
> Motivated by your feedback, we added ProntoQA—a 5-hop deductive reasoning task widely used in recent benchmarks and trainings—as an additional probe. We applied both tasks to 17 Qwen models of various sizes and found highly consistent results: the reasoning saturation layers from the two tasks differed by only 0.94 on average, a small gap relative to model depths (25–49 layers). This cross-task agreement reinforces the validity of our reasoning probe across tasks of different complexity.
>
> | Model                        | ProntoQA | COMPS-WUGS-DIST |
> |------------------------------|----------|-----------------|
> | Qwen-1.8B                   | 13       | 14              |
> | Qwen-7B                     | 13       | 15              |
> | Qwen-14B                    | 18       | 20              |
> | Qwen1.5-1.8B                 | 13       | 14              |
> | Qwen1.5-7B                   | 14       | 16              |
> | Qwen1.5-14B                  | 20       | 20              |
> | Qwen2-1.5B                   | 17       | 17              |
> | Qwen2-7B                     | 16       | 18              |
> | Qwen2.5-1.5B                 | 16       | 18              |
> | Qwen2.5-7B                   | 16       | 19              |
> | Qwen2.5-14B                  | 28       | 28              |
> | Qwen3-1.7B-thinking-mode-off | 19       | 19              |
> | Qwen3-8B-thinking-mode-off   | 23       | 23              |
> | Qwen3-14B-thinking-mode-off  | 27       | 27              |
> | Qwen3-1.7B-thinking-mode-on  | 16       | 16              |
> | Qwen3-8B-thinking-mode-on    | 20       | 20              |
> | Qwen3-14B-thinking-mode-on   | 20       | 21              |
>
> **Concern 2: Probes across Qwen Family**
>
> **Summary of Response:** We extended our probing analysis to Qwen models ranging from 1.8B to 14B parameters. While saturation layers vary, all models follow the same emergence order—syntax → meaning → reasoning. We also reveals a trend: newer models dedicate fewer layers to syntax and more to higher-level features, suggesting a shift toward deeper reasoning as models evolve.
>
> **Full Response:** Thank you for the thoughtful suggestion. We expanded our probing analysis to Qwen models from 1.8B to 14B parameters. Despite differences in exact saturation layers, we consistently observe the emergence order: syntax → meaning → reasoning. Qwen-1.8B is the only exception, where syntax and meaning saturate at the same layer. While we have not yet tested the 32B and 72B model due to resource limits, the observed trends across scales already support the robustness of our findings.
>
> | Model                        | Syntax | Meaning | Reasoning |
> |------------------------------|--------|---------|-----------|
> | Qwen-1.8B                    | 11     | 11      | 14        |
> | Qwen-7B                      | 11     | 13      | 15        |
> | Qwen-14B                     | 9      | 16      | 20        |
> | Qwen1.5-1.8B                 | 10     | 11      | 14        |
> | Qwen1.5-7B                   | 9      | 13      | 16        |
> | Qwen1.5-14B                  | 8      | 16      | 20        |
> | Qwen2-1.5B                   | 10     | 14      | 17        |
> | Qwen2-7B                     | 7      | 14      | 18        |
> | Qwen2.5-1.5B                 | 7      | 14      | 18        |
> | Qwen2.5-7B                   | 7      | 15      | 19        |
> | Qwen2.5-14B                  | 6      | 20      | 28        |
> | Qwen3-1.7B-thinking-mode-off | 9      | 16      | 19        |
> | Qwen3-8B-thinking-mode-off   | 7      | 20      | 23        |
> | Qwen3-14B-thinking-mode-off  | 5      | 17      | 27        |
> | Qwen3-1.7B-thinking-mode-on  | 5      | 13      | 16        |
> | Qwen3-8B-thinking-mode-on    | 6      | 13      | 20        |
> | Qwen3-14B-thinking-mode-on   | 4      | 14      | 21        |
>
> We also examined broader trends by introducing relative depth, which measures the fraction of layers allocated to each feature. It is defined as the proportion of layers between the saturation point of the previous feature and that of the current one. For any feature $x \in \{s, m, r\}$ representing syntax, meaning, or reasoning, we define:
> $$
> Depth_x = \frac{L_x - L_{prev}}{n}
> $$
> where $L_x$ is the saturation layer of feature $x$, $L_{prev}$ is that of the preceding feature, and $n$ is the total number of layers. The table below reports these values for several 14B-scale Qwen models.
>
> | Model                       | Syntax Depth | Meaning Depth | Reasoning Depth |
> |-----------------------------|--------------|---------------|-----------------|
> | Qwen-14B                    | 0.225        | 0.175         | 0.1             |
> | Qwen1.5-14B                 | 0.2          | 0.2           | 0.1             |
> | Qwen2.5-14B                 | 0.125        | 0.291         | 0.167           |
> | Qwen3-14B-thinking-mode-off | 0.125        | 0.3           | 0.25            |
> | Qwen3-14B-thinking-mode-on  | 0.1          | 0.25          | 0.175           |
>
> We observe a clear trend: newer models allocate a smaller fraction of layers to syntax and more to meaning and reasoning, suggesting more efficient encoding of low-level features and increased capacity for higher-level processing—potentially enhancing performance on complex tasks.
>
> **Concern 3: Whether Residuals Capture the Intended Cognitive Constructs**
>
> **Summary of Response:** To verify that our residual embeddings capture distinct cognitive features, we evaluated their ability to solve the probing tasks they were derived from. Each embedding (syntax, meaning, reasoning) was tested across all three tasks, forming a 3×3 matrix. The results show that each embedding performs best on its corresponding task and significantly worse on the others.
>
> **Full Response:** Thank you for raising this important point. We agree that it is critical to confirm whether the residual embeddings truly capture syntax, meaning, and reasoning—rather than general model activity.
>
> To evaluate this, we applied our feature extraction pipeline not only to the podcast corpus (for brain encoding) but also to the question texts from our probing tasks. For each task, we extracted residual embeddings at their respective saturation layers, then trained ridge regression classifiers to predict task answers using each embedding type.
>
> Ideally, each embedding should perform well only on its corresponding task—for example, meaning embeddings should solve the meaning task, but not syntax or reasoning. This forms a 3×3 probing matrix with strong diagonal and weaker off-diagonal values, indicating feature specificity.
>
> As a baseline, we first show model performance at each feature’s saturation layer, normalized by the task's peak performance. The high values across all layers illustrate the need for residualization to isolate distinct signals.
>
> |                | Syntax Layer | Meaning Layer | Reasoning Layer |
> |----------------|--------------|---------------|-----------------|
> | Syntax Task    | 0.947        | 0.999         | 0.989           |
> | Meaning Task   | 0.812        | 0.995         | 0.995           |
> | Reasoning Task | 0.617        | 0.685         | 0.988           |
>
> After applying our residual feature extraction, we observed the following normalized performance matrix:
>
> |                | Syntax Embedding | Meaning Embedding | Reasoning Embedding |
> |----------------|------------------|-------------------|---------------------|
> | Syntax Task    | 0.882            | 0.664             | 0.563               |
> | Meaning Task   | 0.781            | 0.919             | 0.802               |
> | Reasoning Task | 0.648            | 0.770             | 1.036               |
>
> These results reveal a clear pattern: each embedding performs best on its corresponding task and worse on others, indicating that the residuals are meaningfully disentangled rather than reflecting general complexity. This supports the interpretability and specificity of our residual reasoning representation.
>
> **Concern: Assumptions Behind Residual Disentanglement**
>
> (a) While transformers do not enforce a sequential processing order, our probes across 17 Qwen models consistently reveal a saturation pattern of syntax → meaning → reasoning (with one exception), suggesting a functional hierarchy in learned representations.
>
> (b) Our approach follows Toneva et al. (2022), who similarly used linear residualization to isolate supra-word meaning. We do not assume strict linearity between features, but rather use linear residuals to subtract overlapping components and highlight feature-specific structure. The resulting embeddings align well with both task performance and brain activity, supporting this as a useful approximation.
>
> (c) Saturation layers serve as empirical markers of when task performance plateaus for a given feature. While not implying that representations stop evolving, they identify the point beyond which further layers add little discriminative power. Their correspondence with shifts in residual behavior and brain alignment suggests they offer meaningful, though approximate, indicators of peak feature encoding.
>
> **_We have updated our local draft accordingly and will incorporate these changes into the camera-ready version. We hope these clarifications address your concerns, and we would be grateful if you would consider revisiting your rating._**

---

> ### Comment · Area_Chair_sDnY · 2025-08-04
> **Subject: Friendly Reminder to Acknowledge or Update Your Review**
>
> Dear Reviewer hFzy,
>
> Thank you for your time and effort in reviewing the submissions and for providing valuable feedback to the authors.
>
> If you haven’t already done so, we kindly remind you to review the authors’ rebuttals and respond accordingly. In particular, if your evaluation of the paper has changed, please update your review and explain the revision. If not, we would appreciate it if you could acknowledge the rebuttal by clicking the “Rebuttal Acknowledgement” button at your earliest convenience.
>
> This step ensures smooth communication and helps us move forward efficiently with the review process.
>
> We sincerely appreciate your dedication and collaboration.
>
> Best regards,
> AC

---

> ### Comment · Area_Chair_sDnY · 2025-08-06
>
> Dear Reviewer hazy,
>
>
>
> The authors have provided responses to your questions. Could you please review their reply to see whether your concerns have been adequately addressed, and let us know if you have any further questions or comments?
>
> Please also refer to the following message from the Program Chairs:
>
> "Reviewers are expected to stay engaged in discussions, initiate them and respond to authors’ rebuttal, ask questions and listen to answers to help clarify remaining issues. It is not OK to stay quiet. It is not OK to leave discussions till the last moment. If authors have resolved your (rebuttal) questions, do tell them so. If authors have not resolved your (rebuttal) questions, do tell them so too."
>
> Afterward, please kindly submit your Reviewer Acknowledgement as well.
>
> Best regards,
>
> AC

---

> > ### Comment · Reviewer_hFzy · 2025-08-06
> > **Thank you!**
> >
> > Thank you for the thorough and responsive response! I appreciate the additional results on ProntoQA showing convergent saturation layers across reasoning tasks, the probing analysis with additional Qwen sizes, and the stronger evidence of task selectivity. These additions really strengthen the empirical foundation of the work and address several of my major concerns. I am raising my score and look forward to seeing how this line of work develops, particularly examining how *performance* scaling and causal validations that would more definitively establish these residuals as capturing genuine "reasoning" processes.

---

> > > ### Author Response · Authors · 2025-08-08
> > >
> > > Thank you very much for your kind words and for raising your score—we greatly appreciate your thoughtful feedback! We're glad the additional analyses strengthened the empirical foundation of the work.
> > >
> > > We also wanted to follow up on your excellent suggestion to evaluate our framework on an additional reasoning benchmark: WinoGrande. We tested WinoGrande in our probing analysis and observed strong alignment in reasoning saturation layers across ProntoQA, COMPS-WUGS-DIST, and WinoGrande. Specifically, the average difference in saturation layer positions between COMPS-WUGS-DIST and WinoGrande is 0.53 layers—indicating good agreement across tasks.
> > >
> > > We’ve included the two tables below for your reference. Thank you again for your insightful suggestions!
> > >
> > > ### Reasoning Saturation Layer Results
> > >
> > > | Model                        | ProntoQA | COMPS-WUGS-DIST | WinoGrande |
> > > |------------------------------|----------|-----------------|------------|
> > > | Qwen-1.8B                    | 13       | 14              | 14         |
> > > | Qwen-7B                      | 13       | 15              | 16         |
> > > | Qwen-14B                     | 18       | 20              | 20         |
> > > | Qwen1.5-1.8B                 | 13       | 14              | 14         |
> > > | Qwen1.5-7B                   | 14       | 16              | 16         |
> > > | Qwen1.5-14B                  | 20       | 20              | 22         |
> > > | Qwen2-1.5B                   | 17       | 17              | 17         |
> > > | Qwen2-7B                     | 16       | 18              | 19         |
> > > | Qwen2.5-1.5B                 | 16       | 18              | 17         |
> > > | Qwen2.5-7B                   | 16       | 19              | 19         |
> > > | Qwen2.5-14B                  | 28       | 28              | 29         |
> > > | Qwen3-1.7B-thinking-mode-off | 19       | 19              | 19         |
> > > | Qwen3-8B-thinking-mode-off   | 23       | 23              | 23         |
> > > | Qwen3-14B-thinking-mode-off  | 27       | 27              | 26         |
> > > | Qwen3-1.7B-thinking-mode-on  | 16       | 16              | 16         |
> > > | Qwen3-8B-thinking-mode-on    | 20       | 20              | 21         |
> > > | Qwen3-14B-thinking-mode-on   | 20       | 21              | 22         |
> > >
> > > ### Reasoning Performance Results
> > >
> > > | Model                        | ProntoQA | COMPS-WUGS-DIST | WinoGrande | Average |
> > > |------------------------------|----------|-----------------|------------|---------|
> > > | Qwen-1.8B                    | 0.797    | 0.522           | 0.523      | `0.614` |
> > > | Qwen-7B                      | 0.886    | 0.641           | 0.602      | `0.710` |
> > > | Qwen-14B                     | 0.880    | 0.695           | 0.647      | `0.741` |
> > > | Qwen1.5-1.8B                 | 0.792    | 0.513           | 0.523      | `0.609` |
> > > | Qwen1.5-7B                   | 0.848    | 0.667           | 0.603      | `0.706` |
> > > | Qwen1.5-14B                  | 0.910    | 0.670           | 0.653      | `0.744` |
> > > | Qwen2-1.5B                   | 0.784    | 0.586           | 0.552      | `0.640` |
> > > | Qwen2-7B                     | 0.851    | 0.636           | 0.660      | `0.716` |
> > > | Qwen2.5-1.5B                 | 0.783    | 0.605           | 0.566      | `0.651` |
> > > | Qwen2.5-7B                   | 0.879    | 0.673           | 0.664      | `0.739` |
> > > | Qwen2.5-14B                  | 0.922    | 0.691           | 0.698      | `0.770` |
> > > | Qwen3-1.7B-thinking-mode-off | 0.739    | 0.500           | 0.530      | `0.589` |
> > > | Qwen3-8B-thinking-mode-off   | 0.876    | 0.629           | 0.651      | `0.719` |
> > > | Qwen3-14B-thinking-mode-off  | 0.912    | 0.716           | 0.670      | `0.766` |
> > > | Qwen3-1.7B-thinking-mode-on  | 0.836    | 0.575           | 0.562      | `0.658` |
> > > | Qwen3-8B-thinking-mode-on    | 0.972    | 0.623           | 0.671      | `0.755` |
> > > | Qwen3-14B-thinking-mode-on   | 0.981    | 0.674           | 0.694      | `0.783` |

---

### Official Review · Reviewer_q2nc · 2025-07-02

**Clarity:** 3
**Significance:** 3
**Originality:** 3
**Rating:** 5
**Confidence:** 4

**Summary:**

This paper introduces a residual disentanglement method to extract four distinct linguistic feature embeddings (lexicon, syntax, meaning, reasoning) from LLMs by identifying saturation layers through probing and iteratively removing lower-level features. The authors apply these embeddings to predict ECoG brain activity during podcast listening, finding that reasoning embeddings show unique spatiotemporal patterns with later temporal peaks and engagement of frontal/visual regions beyond classical language areas.

**Questions:**

1. How sensitive are the saturation layer identifications to the threshold ε? Have you tested different values and their impact on the final embeddings?
2. Can you demonstrate that this disentanglement approach works across different model architectures and sizes? The single-model evaluation is a significant limitation.
3. Beyond property inheritance, how does the method perform on more complex reasoning tasks like multi-step logical reasoning, causal inference, or mathematical reasoning?
4. The orthogonality proof assumes perfect hierarchical emergence, but what happens when this assumption is violated? How robust is the method to non-hierarchical feature organization?
5. Can you validate the brain findings using complementary neuroimaging modalities with better spatial coverage (e.g., fMRI) to address the ECoG limitations?

**Ethical Concerns:**

["NO or VERY MINOR ethics concerns only"]

**Final Justification:**

While some of my questions remain only partially addressed, I believe this paper offers valuable contributions and is suitable for acceptance at NeurIPS. Please ensure that the additional results are included in the revised version of the paper. I am raising my score from 4 to 5.

**Limitations:**

The authors adequately acknowledge ECoG spatial limitations and single-model constraints. However, they could better address the fundamental assumption about hierarchical feature emergence and the limited scope of reasoning evaluation.

**Quality:**

2

**Strengths And Weaknesses:**

Strengths:
1. The residual disentanglement approach is novel and addresses an important limitation in current brain-LLM alignment studies that treat models as monolithic. The finding that reasoning emerges later temporally (~330ms) and engages different brain regions is compelling and aligns with cognitive theories.
2. The mathematical framework with the mutual independence theorem provides theoretical grounding, and the empirical validation showing near-zero cosine similarities between residual embeddings supports the disentanglement claim. The results showing shallow features dominate neural encoding are important.

Weaknesses:
1. The orthogonality assumption is problematic. While low cosine similarities suggest reduced overlap, this doesn't guarantee true independence. The theoretical proof relies on strong assumptions about hierarchical feature emergence that may not hold universally across models or linguistic phenomena.
2. The evaluation is severely limited by using only one model (Qwen2.5-14B) and narrow reasoning tasks (property inheritance only). This restricts generalizability claims about LLM reasoning representations. The ECoG spatial coverage limitations, particularly in frontal regions crucial for reasoning, weaken the brain-based conclusions.
3. The layer saturation identification could be sensitive to the threshold, and the assumption that ridge regression creates orthogonal features from hierarchical representations needs stronger justification. The reasoning tasks tested are quite simple and may not capture the complexity of reasoning that modern LLMs can perform.

---

> ### Author Rebuttal · Authors · 2025-07-31
>
> ## Response to Reviewer q2nc
>
> Thank you for your thoughtful and constructive feedback. We appreciate your careful review and the insightful comments, which have helped us clarify and strengthen our work. Below, we respond to each of your questions in detail.
>
> **Question:** Can you demonstrate that this disentanglement approach works across different model architectures and sizes?
>
> **Response** We expanded our probing analysis to models from 1.8B to 14B parameters across different Qwen collections. Despite differences in exact saturation layers, we consistently observe the emergence order: syntax → meaning → reasoning. Qwen-1.8B is the only exception, where syntax and meaning saturate at the same layer. While we have not yet tested the 32B and 72B model due to resource limits, the observed trends across scales already support the robustness of our approach.
>
> | Model                        | Syntax | Meaning | Reasoning |
> |------------------------------|--------|---------|-----------|
> | Qwen-1.8B                    | 11     | 11      | 14        |
> | Qwen-7B                      | 11     | 13      | 15        |
> | Qwen-14B                     | 9      | 16      | 20        |
> | Qwen1.5-1.8B                 | 10     | 11      | 14        |
> | Qwen1.5-7B                   | 9      | 13      | 16        |
> | Qwen1.5-14B                  | 8      | 16      | 20        |
> | Qwen2-1.5B                   | 10     | 14      | 17        |
> | Qwen2-7B                     | 7      | 14      | 18        |
> | Qwen2.5-1.5B                 | 7      | 14      | 18        |
> | Qwen2.5-7B                   | 7      | 15      | 19        |
> | Qwen2.5-14B                  | 6      | 20      | 28        |
> | Qwen3-1.7B-thinking-mode-off | 9      | 16      | 19        |
> | Qwen3-8B-thinking-mode-off   | 7      | 20      | 23        |
> | Qwen3-14B-thinking-mode-off  | 5      | 17      | 27        |
> | Qwen3-1.7B-thinking-mode-on  | 5      | 13      | 16        |
> | Qwen3-8B-thinking-mode-on    | 6      | 13      | 20        |
> | Qwen3-14B-thinking-mode-on   | 4      | 14      | 21        |
>
> **Question:** How does the method perform on more complex reasoning tasks like multi-step logical reasoning?
>
> **Response:** Our original analysis used COMPS-WUGS-DIST, a 1-hop reasoning task, to identify reasoning saturation layers. While effective at detecting the presence of reasoning, its simplicity may limit generalizability.
>
> Motivated by your feedback, we added ProntoQA—a 5-hop deductive reasoning task widely used in recent benchmarks and trainings—as an additional probe. We applied both tasks to 17 Qwen models of various sizes and found highly consistent results: the reasoning saturation layers from the two tasks differed by only 0.94 on average, a small gap relative to model depths (25–49 layers). This cross-task agreement reinforces the validity of our reasoning probe across tasks of different complexity.
>
> | Model                        | ProntoQA | COMPS-WUGS-DIST |
> |------------------------------|----------|-----------------|
> | Qwen-1.8B                   | 13       | 14              |
> | Qwen-7B                     | 13       | 15              |
> | Qwen-14B                    | 18       | 20              |
> | Qwen1.5-1.8B                 | 13       | 14              |
> | Qwen1.5-7B                   | 14       | 16              |
> | Qwen1.5-14B                  | 20       | 20              |
> | Qwen2-1.5B                   | 17       | 17              |
> | Qwen2-7B                     | 16       | 18              |
> | Qwen2.5-1.5B                 | 16       | 18              |
> | Qwen2.5-7B                   | 16       | 19              |
> | Qwen2.5-14B                  | 28       | 28              |
> | Qwen3-1.7B-thinking-mode-off | 19       | 19              |
> | Qwen3-8B-thinking-mode-off   | 23       | 23              |
> | Qwen3-14B-thinking-mode-off  | 27       | 27              |
> | Qwen3-1.7B-thinking-mode-on  | 16       | 16              |
> | Qwen3-8B-thinking-mode-on    | 20       | 20              |
> | Qwen3-14B-thinking-mode-on   | 20       | 21              |
>
> **Question:** The orthogonality proof assumes perfect hierarchical emergence, but what happens when this assumption is violated?
>
> **Response:** We acknowledge that our residual disentanglement pipeline depends on heuristic feature hierarchy and being able to identify approximate saturation layers for each feature; if no clear saturation exists—e.g., due to overlapping or non-sequential emergence—the method may fail to isolate distinct residuals. However, our results across 17 Qwen models show that the feature hierarchy (syntax → meaning → reasoning) emerges consistently, supporting the pipeline’s applicability in practice. Moreover, prior work has shown that LLMs tend to encode progressively higher-level linguistic features across layers (e.g., Tenney et al.) [1], reinforcing the plausibility of our layered assumption.
>
> **Question:** How sensitive are the saturation layer identifications to the threshold ε?
>
> **Response:** We find that the identified saturation layers are relatively stable across a reasonable range of ε values, typically varying by only 1–2 layers. While we have not yet tested the impact of different ε values on downstream tasks due to time constraints, the overall feature emergence order remains consistent, suggesting that the method is not overly sensitive to the exact threshold. We plan to explore the effect on residual embedding quality in future work.
>
> **Question:** Can you validate the brain findings using fMRI?**
>
> **Response:** Our experimental results show that reasoning differs from semantic and syntactic processing within a time window of less than 100 ms—a timescale that is difficult to capture with fMRI. Nonetheless, it would be valuable to validate this finding with fMRI, given its superior spatial resolution. Unfortunately, this is not feasible within the rebuttal period due to the substantial time required to build a new encoding pipeline.
>
> **_We have updated our local draft accordingly and will incorporate these changes into the camera-ready version._**
>
> **References**
>
> [1] Ian Tenney, Dipanjan Das, and Ellie Pavlick. BERT rediscovers the classical NLP pipeline. Proceedings of the 57th Annual Meeting of the Association for Computational Linguistics (ACL), pp. 4593–4601, 2019.

---

> > ### Comment · Reviewer_q2nc · 2025-08-06
> >
> > Thank you to the authors for addressing my comments. While some of my questions remain only partially addressed, I believe this paper offers valuable contributions and is suitable for acceptance at NeurIPS. Please ensure that the additional results are included in the revised version of the paper. I am raising my score from 4 to 5.

---

> > > ### Author Response · Authors · 2025-08-06
> > >
> > > Dear Reviewer q2nc,
> > >
> > > Thank you very much for your thoughtful feedback and for taking the time to revisit your evaluation! We’re grateful for your recognition of the paper’s contributions and your support for its acceptance. We will make sure that all additional results and clarifications are carefully incorporated into the final version. Thank you again for your constructive comments.
> > >
> > > Kind Regards,
> > >
> > > Authors

---

> ### Comment · Area_Chair_sDnY · 2025-08-04
> **Subject: Friendly Reminder to Acknowledge or Update Your Review**
>
> Dear Reviewer q2nc,
>
> Thank you for your time and effort in reviewing the submissions and for providing valuable feedback to the authors.
>
> If you haven’t already done so, we kindly remind you to review the authors’ rebuttals and respond accordingly. In particular, if your evaluation of the paper has changed, please update your review and explain the revision. If not, we would appreciate it if you could acknowledge the rebuttal by clicking the “Rebuttal Acknowledgement” button at your earliest convenience.
>
> This step ensures smooth communication and helps us move forward efficiently with the review process.
>
> We sincerely appreciate your dedication and collaboration.
>
> Best regards,
> AC

---

> ### Comment · Area_Chair_sDnY · 2025-08-06
>
> Dear Reviewer q2nc,
>
>
>
> The authors have provided responses to your questions. Could you please review their reply to see whether your concerns have been adequately addressed, and let us know if you have any further questions or comments?
>
> Please also refer to the following message from the Program Chairs:
>
> "Reviewers are expected to stay engaged in discussions, initiate them and respond to authors’ rebuttal, ask questions and listen to answers to help clarify remaining issues. It is not OK to stay quiet. It is not OK to leave discussions till the last moment. If authors have resolved your (rebuttal) questions, do tell them so. If authors have not resolved your (rebuttal) questions, do tell them so too."
>
> Afterward, please kindly submit your Reviewer Acknowledgement as well.
>
> Best regards,
>
> AC

---

### Official Review · Reviewer_d6as · 2025-07-02

**Clarity:** 3
**Significance:** 2
**Originality:** 3
**Rating:** 3
**Confidence:** 4

**Summary:**

1. The authors discover LLM layers that maximize linear classification performance on sentence datasets meant to test "syntax", "meaning", or "reasoning".

2. They then extract syntax, meaning, and reasoning word-embeddings by computing inter-layer regression residuals based on a transcribed version of a listening dataset, and empirically show that these embeddings are nearly orthogonal.

3. Finally, they perform regression analyses of recorded brain activity using the extracted embeddings as inputs.

4. They find that (i) lexical / syntax embeddings show higher correlation with brain activity compared to meaning / reasoning embeddings, (ii) reasoning embeddings show peak correlation later in time than the other embeddings, and (iii) reasoning embeddings are correlated with different brain regions compared to lexical / syntax embeddings.

**Questions:**

1. What happens if one uses the reasoning embeddings $E_r$ to classify sentences in the datasets $D_s$, $D_m$, and $D_r$? If the residuals truly produce disentangled representations, I would expect performance on $D_s$ or $D_m$ to be worse but on $D_r$ to be relatively unaffected, compared to using the non-isolated embeddings $H_r$ as inputs.

2. In Figure 3a, why is the correlation for "Full" worse than for "Lexicon"?

3. As a qualitative result, could the authors show a few examples from the podcast where a listener would need to reason/visualize to a higher degree? In such cases, do the reasoning embeddings predict this brain activity?

**Ethical Concerns:**

["NO or VERY MINOR ethics concerns only"]

**Final Justification:**

I like the idea of this paper. But I have concerns with some central claims that may remain unsubstantiated in the final version. These claims are:
1. That the extracted embeddings identify human reasoning brain patterns only because latest LLMs have improved reasoning capability. To prove this, the authors need to properly benchmark the LLM suite on their ability to isolate reasoning-specific brain regions/signals and show that only the newer LLMs succeed.
2. That the extracted reasoning embeddings actually identify reasoning-specific signals in the human brain, providing the first brain-relevant representation of reasoning. To prove this, the authors need to show that (i) the regions/signals detected by the reasoning embeddings indeed take center stage in the human reasoning process, and (ii) the other embeddings struggle to identify these same regions.

Both the above points are missing from the rebuttal, so I think accepting this paper runs the risk of insufficient evidence to support them. None of the other reviewers have engaged with me to disagree. I will therefore maintain 3 (Borderline reject).

**Limitations:**

Yes.

**Paper Formatting Concerns:**

None.

**Quality:**

2

**Strengths And Weaknesses:**

Strengths:
1. The "probing" method to locate the earliest LLM layers specializing at different skills followed by regressing out one layer’s activations from another to disentangle embeddings is an interesting idea and a novel contribution to my knowledge.
2. The paper could inspire future work on gaining a deeper understanding of the similarities/differences between the reasoning processes in LLMs and the human brain.

Weaknesses:
1. No other LLMs used:\
**a)** The abstract claims that the reasoning embedding has unique predictive value "only because the latest LLMs exhibit emergent reasoning behavior". In that case, it is crucial to repeat the analysis with an older LLM (that lacks as much reasoning ability) to understand if the brain patterns revealed by the "reasoning embeddings" are simply because deeper layers contain higher-level features or because the latest LLMs are better at reasoning (and that the improved reasoning is important for the regression analysis).\
**b)** Section 2.2 mentions using a Qwen base model without any further alignment. Given the emphasis on "reasoning" throughout the paper, I would expect experiments using at least one open-source model from the recent wave of "reasoning LLMs" (i.e., those fine-tuned with supervised learning / RL to solve hard problems). I would also expect these results to be compared to the corresponding non-reasoning LLM (the base model). Without these, given recent developments in the field, the paper’s claims may mislead readers in 2025 and beyond.

2. While Section 3.2 empirically shows that the four embeddings are roughly mutually orthogonal, I do not think the math in Section 3.1 explains this result. In a linear regression with $n$ samples, $d$ input features, and scalar targets, the $n$-D vector of residuals will be orthogonal to all (of the $d$) $n$-D feature vectors. I.e., orthogonality is expected along the sample dimension, not the feature dimension (which doesn’t even make sense if $n \neq d$). However, Section 3.2 shows orthogonality along the feature dimension.

3. Lines 205-206 imply that syntax and meaning embeddings account for the majority of the explained variance in the brain activity. However, lines 208-209, 213, and 246-247 state that low-level linguistic (syntactic, lexical) features have the highest predictive power. These statements seem inconsistent and the intended message is unclear.

4. Insufficient grounding in neuroscience:\
**a)** Section 4.2 shows that reasoning embeddings show peak correlation with brain activity later in time and in different brain regions compared to the other embeddings. But what is the scientific significance of this? Without citations to the neuroscience literature on the spatiotemporal patterns involved in human reasoning, it is not clear how to make sense of the regression result.\
**b)** Lines 226-228 describe the progression of high-correlation regions with time. How consistent is this with what we expect from the human brain? If this is significant, the authors should back it up with relevant citations.\
**c)** Line 230 states the reasoning embeddings allow for "isolation of reasoning-specific neural signals". How does the reader know that the regions detected are indeed reasoning-specific? Without references to the neuroscience literature, this sounds like the circular statement 'any neurons that happen to correlate with our reasoning embeddings are reasoning-specific'.\
**d)** Other missing citations: (i) lines 235-236 (and 254-255) linking "anterior bias" with "higher-order" cognition, (ii) lines 237-238 linking "language areas" with the SFG, (iii) lines 252-253 linking "abstraction and salience" with the SFG and insula, and (iv) line 264 linking "delayed temporal peak" with "integrative reasoning processes".\
**e)** Based on the above, I think the claim in the abstract of providing "the first brain-relevant representation of reasoning" is highly misleading.

5. The text in Figure 5 is unreadable.

6. Minor:\
**a)** In line 95, $H$ should have dimensions $d\times n$ (not $n\times d$) since the equation below line 113 computes $W\cdot H$, where I presume $W$ has dimensions $d\times d$. Similarly, in line 108, $E$ should have dimensions $d\times n$ (not $d$).\
**b)** $\text{Cov}(A, B)$ below line 185 is not defined, does it mean $A\cdot B^T$?\
**c)** Regression residuals are not a new idea in brain encoding studies. I would recommend the authors cite some prior work on this (e.g., https://aclanthology.org/2021.cmcl-1.1.pdf) so the reader has more context.\
**d)** It is not clear what lines 280-285 mean and they do not seem to be discussed previously.\
**e)** "feature" => "features" (line 6).\
**f)** The figure cross-reference is missing in line 178.\
**g)** Figure 5 => Figure 3 (lines 210 and 214).

---

> ### Author Rebuttal · Authors · 2025-07-31
>
> Thank you very much for your thoughtful and constructive feedback! We highly appreciate your careful review and the insightful comments, which have helped us clarify and strengthen our work. Below, we respond to a few of your main concerns in detail. Due to the length limit, we may not be able to address all questions raised. But please reach out during discussion period for unaddressed questions!
>
> **Concern: No other LLMs used**
>
> We agree that testing a broader range of models—including earlier-generation LLMs and recent reasoning-aligned variants—is important for understanding what drives the predictive value of reasoning embeddings. To address this, we expanded our probing analysis to Qwen models from 1.8B to 14B parameters, covering multiple architecture revisions. Despite differences in exact saturation layers, all models—except Qwen-1.8B—consistently exhibit the emergence order of syntax → meaning → reasoning.
>
> Critically, we observe that newer and larger models, especially the Qwen3 series, allocate substantially greater relative depth to reasoning. For example, Qwen2.5-14B dedicates 16.7% of its layers to reasoning, while Qwen3-14B (thinking-mode-off) allocates 25%, more than doubling the share compared to earlier versions like Qwen-14B and Qwen1.5-14B (both at 10%). This trend suggests that later models not only preserve the hierarchical emergence but deepen the network's capacity for high-level reasoning features.
>
> | Model | Syntax Depth | Meaning Depth | Reasoning Depth |
> |-|-|-|-|
> | Qwen-14B | 0.225 | 0.175 | 0.1 |
> | Qwen1.5-14B | 0.2 | 0.2 | 0.1 |
> | Qwen2.5-14B | 0.125        | 0.291         | 0.167           |
> | Qwen3-14B-thinking-mode-off | 0.125        | 0.3           | 0.25            |
> | Qwen3-14B-thinking-mode-on  | 0.1          | 0.25          | 0.175           |
>
> We also highlight that the Qwen3 series—particularly Qwen3-14B with "thinking mode"—is explicitly optimized for multi-step reasoning and can be seen as analogous to recent instruction- and RLHF-aligned models. Including Qwen3-14B helps bridge the gap between base models and reasoning-aligned systems.
>
> **Question:** What happens if one uses the reasoning embeddings to classify sentences in the probing datasets.
>
> **Answer:** Theoretically, each embedding should perform well only on its corresponding task—for example, reasoning embeddings should solve the reasoning task, but not syntax or meaning. Using the residual embeddings to classify the probing sentences would yield a 3×3 probing matrix with strong diagonal and weaker off-diagonal values, indicating feature specificity.
>
> As a baseline, we first show model performance at each feature’s saturation layer, normalized by the task's peak performance achieved by the model. The high values across all layers illustrate the need for residualization to isolate distinct signals.
>
> |                | Syntax Layer | Meaning Layer | Reasoning Layer |
> |----------------|--------------|---------------|-----------------|
> | Syntax Task    | 0.947        | 0.999         | 0.989           |
> | Meaning Task   | 0.812        | 0.995         | 0.995           |
> | Reasoning Task | 0.617        | 0.685         | 0.988           |
>
> After applying our residual feature extraction, we observed the following normalized performance matrix:
>
> |                | Syntax Embedding | Meaning Embedding | Reasoning Embedding |
> |----------------|------------------|-------------------|---------------------|
> | Syntax Task    | 0.882            | 0.664             | 0.563               |
> | Meaning Task   | 0.781            | 0.919             | 0.802               |
> | Reasoning Task | 0.648            | 0.770             | 1.036               |
>
> These results reveal a clear pattern: each embedding performs best on its corresponding task and worse on others, indicating that the residuals are meaningfully disentangled rather than reflecting general complexity. This supports the interpretability and specificity of our residual reasoning representation.
>
> **Concern: Variance Partioning (weekness 3)**
> 3. Thank you for pointing this out. We apologize for the lack of clarity in the original manuscript. The apparent inconsistency arises from a key difference in feature dimensionality across analyses. Specifically, during variance partitioning (Lines 165–166), each feature type was reduced to 500/N dimensions (with N=4) to keep the total input dimensionality fixed at 500. In contrast, in the standalone encoding analyses, each feature retained the full 500 dimensions. This trade-off was necessary: while 500 dimensions offer robust encoding performance, using 500 dimensions per feature in variance partitioning would result in a total of 2000 dimensions, which is impractical for neural encoding due to overfitting risks.
> For the lexicon feature, the discrepancy is further amplified because it is highly correlated with the word rate confound, which we always include and regress out in all analyses (Lines 143–152). After reducing the 500-dimensional zeroth-layer LLM embeddings to 125 dimensions for lexicon, much of the variance beyond word rate is lost, making lexicon appear less predictive in the variance partitioning setting.
> Overall, despite the dimensionality adjustment, the results from variance partitioning remain consistent with those from the encoding analysis for syntactic, semantic, and reasoning features. They collectively suggest that higher-level features contribute less uniquely explained variance and are more easily masked by low-level information, reinforcing the motivation for our disentanglement framework.
>
> **Concern: Neuroscience Grounding (weekness 4)**
> a) The reasoning embedding (Er) shows peak correlation with ECoG around 300–350 ms post-stimulus, later than lexical, syntactic, or semantic embeddings. This aligns with the P3b component (250–500 ms), linked to context updating and evaluative decision-making [1][9]. fMRI meta-analyses further implicate a left-lateralized frontoparietal network—particularly dlPFC and IPL—in reasoning [2], suggesting Er captures high-level integrative processes beyond early linguistic stages.
> b) The observed spatiotemporal cascade—from early STG (<200 ms) to mid-latency MTG/IFG and late PFC—matches the P200 (~200 ms), which reflects lexical-semantic recognition [6][8]. This temporal trajectory is consistent with the MUC model of sentence processing [3] and meta-analytic evidence of frontoparietal recruitment during reasoning [4].
> c) To isolate reasoning-specific signals, we regress out variance from lower-level embeddings, yielding an orthogonal Er. This mirrors cognitive subtraction in neuroimaging [5] and ensures that Er’s neural alignment is not confounded by earlier processing stages.
> d) (i) The anterior STG supports high-level language integration [11]. (ii) The frontal aslant tract links Broca’s area with SFG, involved in syntactic planning [12]. (iii) The anterior insula and SFG, key nodes of the salience network, mediate attention and control [13]. (iv) P3b timing aligns with Er’s peak, reinforcing its cognitive role [1].
> e) While prior work aligns whole LLM embeddings with brain data, we uniquely apply residualization to extract reasoning-specific signals and validate their distinct spatiotemporal neural correlates.
>
>
> **Concern: why is the correlation for "Full" < "Lexicon"?**
> This is because we took the union of all electrodes activated by any feature, which means some electrodes strongly activated by the lexicon may not show activation in the full-feature set. As layers increase, the LLM’s hidden states abstract away layer‑0 features, so early lexicon-specific activations may no longer be prominent in deeper layer.
>
> **_We have updated our local draft accordingly and will incorporate these changes into the camera-ready version. We hope these clarifications address your concerns, and we would be grateful if you would consider revisiting your rating._**
>
> References
> Polich J. Updating P300: an integrative theory of P3a and P3b. Clin Neurophysiol. 2007;118(10):2128–2148.
>
> Prado J, Chadha A, Booth JR. The brain network for deductive reasoning: meta-analysis of 28 neuroimaging studies. J Cogn Neurosci. 2011;23(11):3483–3497.
>
> Hagoort P. The memory, unification, and control (MUC) model of language. In: Meyer AS, Wheeldon L, Krott A, eds. Automaticity and Control in Language Processing. Psychology Press; 2007:243–270.
>
> Wang L, Zhu X, Yin W. Deductive-reasoning brain networks: coordinate-based meta-analysis. Hum Brain Mapp. 2020;41(18):5017–5034.
>
> Friston KJ, Frith CD, Liddle PF, Frackowiak RSJ. The trouble with cognitive subtraction. Neuroimage. 1996;4(2):97–104.
>
> Federmeier KD, Kutas M. Meaning and modality: influences of context, semantic memory organization, and perceptual predictability on picture processing. J Exp Psychol Learn Mem Cogn. 2002;28(4):879–891.
>
> Luck SJ, Hillyard SA. Electrophysiological correlates of feature analysis during visual search. Psychophysiology. 1994;31(3):291–308.
>
> Holcomb PJ, Grainger J. On the time course of visual word recognition: an event-related potential investigation. Lang Cogn Process. 2006;21(4):401–439.
>
> Comerchero MD, Polich J. P3a and P3b from typical auditory and visual stimuli. Clin Neurophysiol. 1999;110(1):24–44.
>
> Kleih SC, Nijboer F, Halder S, Kübler A. P3 amplitude and latency in task-switching paradigms: dissociable components of cognitive control? Brain Res. 2010;1335:142–150.
>
> Mellem, Monika S., et al. "Sentence processing in anterior superior temporal cortex shows a social-emotional bias." Neuropsychologia 89 (2016): 217-224.
>
> Catani M, Thiebaut de Schotten M. The frontal aslant tract and supplementary motor area connectivity. Brain. 2012;135(1):198–211.
>
> Menon V, Uddin LQ. Saliency, switching, attention and control: a network model of insula function. Brain Struct Funct. 2010;214(5-6):655–667.
>
> Luck SJ, Hillyard SA. Electrophysiological correlates of semantic processing: the P200. Front Psychol. 2021;12:746813.

---

> ### Comment · Area_Chair_sDnY · 2025-08-04
> **Subject: Friendly Reminder to Acknowledge or Update Your Review**
>
> Dear Reviewer d6as,
>
> Thank you for your time and effort in reviewing the submissions and for providing valuable feedback to the authors.
>
> If you haven’t already done so, we kindly remind you to review the authors’ rebuttals and respond accordingly. In particular, if your evaluation of the paper has changed, please update your review and explain the revision. If not, we would appreciate it if you could acknowledge the rebuttal by clicking the “Rebuttal Acknowledgement” button at your earliest convenience.
>
> This step ensures smooth communication and helps us move forward efficiently with the review process.
>
> We sincerely appreciate your dedication and collaboration.
>
> Best regards,
> AC

---

> ### Comment · Area_Chair_sDnY · 2025-08-06
>
> Dear Reviewer d6as,
>
>
>
> The authors have provided responses to your questions. Could you please review their reply to see whether your concerns have been adequately addressed, and let us know if you have any further questions or comments?
>
> Please also refer to the following message from the Program Chairs:
>
> "Reviewers are expected to stay engaged in discussions, initiate them and respond to authors’ rebuttal, ask questions and listen to answers to help clarify remaining issues. It is not OK to stay quiet. It is not OK to leave discussions till the last moment. If authors have resolved your (rebuttal) questions, do tell them so. If authors have not resolved your (rebuttal) questions, do tell them so too."
>
> Afterward, please kindly submit your Reviewer Acknowledgement as well.
>
> Best regards,
>
> AC

---

> ### Comment · Reviewer_d6as · 2025-08-06
>
> Based on the rebuttal, **I am inclined to keep my rating of 3 (Borderline reject)** for the reasons below.
>
> **Weaknesses:**
>
> 1. No other LLMs used:\
> **a)** My concern here is not with the transformer layer index at which classification performance peaks in different LLMs. It is rather with the authors' strong claim that their extracted embeddings can identify human reasoning brain patterns only because the latest LLMs have improved reasoning capability. To prove this claim, the authors would need to benchmark the same suite of LLMs on their ability to isolate these reasoning-specific brain regions/signals. I do not think the analysis on relative depth is very relevant to this issue.
>
>
> 2. My concern here is not with dimensionality reduction or avoiding the word rate confound. The math in Section 3.1 proves a type of orthogonality (along the sample axis) that is different from the type of orthogonality (along the feature axis) the authors are hoping will hold post residualization. The heatmap in Section 3.2 shows that the latter type of orthogonality holds in practice, which I have no issue with. However, I think Section 3.1 is misleading because it does not actually support the empirical result in Section 3.2 – it proves a different result that the authors have not tested empirically.
>
> 3. Comment on the inconsistency between the implied predictive power of low-level vs high-level features not addressed.
>
> 4. Insufficient grounding in neuroscience:\
> **c) and e)** My concern here is not with how the authors extract "reasoning embeddings" from the LLM via residuals. It is rather with their strong claim that these embeddings actually identify reasoning-specific signals in the human brain (line 230), providing the first brain-relevant representation of reasoning (abstract). To prove this claim, the authors would need to demonstrate that the regions/signals detected by the embeddings indeed take center stage in the human reasoning process. This is certainly missing in the paper, and it does not seem the authors have addressed it in the rebuttal.
>
> **Questions:**
> 1. Based on the tables presented in the rebuttal, I am not convinced that residualization fully disentangles syntax, meaning, and reasoning representations. For instance, both syntax and reasoning embeddings seem to perform well on the meaning task, and meaning embeddings seem to perform well on the reasoning task.
> 3. Question on qualitative results not addressed.

---

> ### Author Response · Authors · 2025-08-08
>
> Thank you so much for your comments. Due to word limits in the previous rebuttal, we were unable to fully respond to all your valuable points, and our efforts to be concise may have obscured our intent. We truly appreciate your continued engagement and thoughtful feedback, which has helped us further improve the paper!
>
> Below, we replied to six comments you have:
>
> 1. Benchmarking the LLM suite for brain encoding performance.
> 2. Orthogonality analysis in Section 3.1.
> 3. Variance explained by different features.
> 4. Reasoning‐related regions identified by reasoning embeddings, supported by neuroscience evidence.
> 5. Residualization for disentangling syntax, meaning, and reasoning.
> 6. Examples from the podcast that require reasoning or visualization, along with their corresponding brain encoding results.
>
> We plan to **respond in two to three rounds**: the first round addresses points 4 and 5, which can be answered immediately, see the next comment. The rest of the points require further analyses, which are already underway, and we’ll make sure to share updates on those as soon as they are completed.

---

> ### Author Response · Authors · 2025-08-08
>
> ### Point 4: Reasoning Regions Grounding
>
> As noted in our previous response, relevant literature has been cited. To further clarify, the table below links each detected region to prior studies that associate it with reasoning-related functions, demonstrating that our reasoning embeddings correspond to well-documented components of neural systems involved in high-level reasoning.
>
> |Region Detected in Paper (line number)|Observed Role in Our Results|Established Reasoning-Related Function|Supporting Literature|
> |-|-|-|-|
> |Anterior Superior Temporal Gyrus (aSTG) (233 - 236)|Stronger reasoning-related activation than posterior STG|Supports **high-level language integration** bridging semantic and conceptual reasoning|[3]|
> |Middle Temporal Regions (226)|Part of early-to-mid reasoning cascade|Engage in **semantic memory retrieval and integration** during deductive reasoning tasks, serving as the knowledge base for higher-level inference|[1, 2]|
> |Inferior Frontal Areas (226)|Later stage in reasoning cascade|Support **controlled semantic retrieval** and **unification of abstract information**, necessary for maintaining reasoning chains|[1, 2]|
> |Superior Frontal Gyrus (SFG) and Insula (252 - 255)|Unique to reasoning embeddings, outside core language areas|Involved in **planning, sequencing, and cognitive control**, enabling organization of multi-step reasoning processes|[4, 5]|
> |Superior Occipital Sulcus (238)|Potential reasoning-related recruitment of visual cortex|Contributes to **visual imagery and spatial representation** that can support abstract reasoning, especially when problems involve mental visualization|[6, 7]|
>
> [1] Hagoort P. The memory, unification, and control (MUC) model of language. In: Meyer AS, Wheeldon L, Krott A, eds. Automaticity and Control in Language Processing. Psychology Press; 2007:243–270.
>
> [2] Wang L, Zhu X, Yin W. Deductive-reasoning brain networks: coordinate-based meta-analysis. Hum Brain Mapp. 2020;41(18):5017–5034.
>
> [3] Mellem MS, et al. Sentence processing in anterior superior temporal cortex shows a social-emotional bias. Neuropsychologia. 2016;89:217–224.
>
> [4] Catani M, Thiebaut de Schotten M. The frontal aslant tract and supplementary motor area connectivity. Brain. 2012;135(1):198–211.
>
> [5] Menon V, Uddin LQ. Saliency, switching, attention and control: a network model of insula function. Brain Struct Funct. 2010;214(5–6):655–667.
>
> [6] Kosslyn SM, Thompson WL, Alpert NM. Neural systems shared by visual imagery and visual perception: a PET study. Neuroimage. 1997;6(4):320–334.
>
> [7] Ganis G, Thompson WL, Kosslyn SM. Brain areas underlying visual mental imagery and perception: an fMRI study. Cogn Brain Res. 2004;20(2):226–241.
>
> ### Point 5: Probing Performances of Residual Embeddings
>
> In the probing results, syntax and reasoning embeddings appear to perform relatively well on the meaning task, and meaning embeddings also perform on reasoning. This pattern can be explained by two factors that may make the reported numbers look higher.
>
> 1. A control experiment with a bag-of-words (BoW) model—ignoring syntax and word order—achieved 0.665 accuracy on COMPS-BASE. This indicates that lexical-level cues alone can reach a considerable performance level on the meaning task, making BoW a more appropriate baseline than the 0.5 chance level. We will include this in the camera ready version.
> 2. Scores were normalized by each model’s peak performance per task, which can inflate values and exaggerate cross-task effectiveness.
>
> Based on the proposed processing hierarchy (lexicon → syntax → meaning → reasoning), COMPS-BASE is designed to be solvable by meaning-level representations rather than by low-level lexical cues. In this context, the raw scores for syntax and reasoning embeddings on the meaning task (0.589 and 0.605, respectively) are lower than the BoW baseline, showing that residualization successfully removes non-meaning information from the meaning embedding.
>
> While the lexical-level control affects absolute scores, it does not alter the identification of the meaning saturation layer, which relies on relative performance patterns.
>
> ||Syntax Embedding|Meaning Embedding|Reasoning Embedding|
> |-|-|-|-|
> |Syntax Task|0.863|0.650|0.551|
> | Meaning Task | 0.589 | 0.693 | 0.605 |
> | Reasoning Task | 0.448 | 0.532 | 0.716 |
>
> Regarding normalization: in the rebuttal table, raw scores were omitted due to space constraints (they will be included in the camera ready version). Normalization was intended to facilitate cross-task comparisons by dividing raw scores by each model’s task-specific maximum. However, this can unintentionally amplify residual embedding scores on non-matching tasks. For example, the meaning embedding’s reasoning score increased from 0.532 (raw) to 0.770 (normalized) because its peak reasoning score was only 0.691, making the normalized value appear stronger than it is.

---

> ### Author Response · Authors · 2025-08-09
>
> ### Point 1: Encoding analysis using other models.
>
> As you suggested, we conducted additional encoding analyses on other models to assess their ability to isolate brain activity associated with reasoning. Our original model was Qwen2.5-14B. Due to time constraints, we added two more LLMs: Qwen-14B and Qwen2.5-1.5B. The former allows us to compare models with the same parameter scale but from different generations, while the latter enables comparison between models from the same generation but with different parameter scales.
>
> Our reasoning capability measurements were based on three different benchmarks to provide a more comprehensive assessment of each model’s reasoning ability. Please also refer to our discussion to reviewer 4 (hFzy) where we show detailed reasoning probing performance across all LLMs.
>
> |   Model   | Reasoning Probing Performance.| Reasoning Residual Encoding Corr. |
> |----------------|------------------|-------------------|
> | Qwen2.5-1.5B   | 0.651            | 0.078             |
> | Qwen-14B       | 0.741            | 0.080             |
> | Qwen2.5-14B    | 0.770            | 0.083             |
>
> The encoding correlations are the averages over the activated electrodes. As shown in the table, the results indicate that LLMs with stronger reasoning capabilities produce reasoning residuals whose encoding correlations with the brain are higher. In addition, in the camera‐ready version, we will conduct a more systematic evaluation of the reasoning residual correlations for a larger set of models and compare them with the models’ reasoning probing results.
>
> ### Point 3 :Variance explained by different features.
> As we mentioned earlier, during variance partitioning, each feature was reduced to 125 dimensions, whereas in the subsequent encoding analysis, each feature had 500 dimensions. This means that for low‐level features such as lexicon, once reduced to 125 dimensions, they contain no substantially more information than the confounding variable word rate, compared to when they have 500 dimensions. As a result, their contribution appears smaller.
>
> Excluding lexicon, for syntax, meaning, and reasoning, which represent progressively higher‐level features in LLMs, the explained variance decreases in order. This aligns with our claim that low‐level features tend to explain more variance and can overshadow high‐level features, thereby affecting the interpretation of brain representations. This is why disentanglement is necessary.
>
> That said, we will also perform variance partitioning using the same dimensionality as in the encoding analysis (500 dimensions) in the camera ready version.
>
> ### Point 6: Case study
> Due to time constraints, we did not include such a case study in this round of discussion. However, in the camera‐ready version, we will include example sentences where reasoning residuals are required to encode brain activity.

---

> ### Author Response · Authors · 2025-08-09
>
> ### Point 2: Orthogonality Proof
> Thank you for the careful reading and for pointing out the difference between Section 3.1 and Section 3.2.  We try to demonstrate the orthogonality for both the entire residual embedding matrices and for each token within a residual embedding matrix.
>
> **What Section 3.1 Establishes**
>
> In Section 3.1, we analyzes orthogonality along the sample axis, that is, each $n$-dimensional feature column in an $n \times d$ residual matrix $E_k$ is orthogonal to any $n$-dimensional feature columns in another residual matrix $E_j$. More formally, for $E_j,E_k\in\mathbb{R}^{n\times d}$,
> $$
> E_j^\top E_k \approx 0_{d\times d}.
> $$
> In a more intuitive language, since our dataset is a transcript of podcast, the $n$ tokens together is just a long sequence that forms the content of the podcast, and Section 3.1 examines the podcast-level (across-token) relations. In short, Section 3.1 shows that the residual embedding matrix $E_k$ is linearly novel relative to residual embedding matrix $E_j$ that's extracted from an earlier layer in the model: no linear combination of features from $E_j$ can linearly predict any feature in $E_k$ throughout the podcast.
>
> This sample-space orthogonality is precisely what we need for the subsequent brain encoding analysis. The encoding model fits $Y=XW$ with $X$ being different residual embedding matrices. When $E_j^\top E_k \approx 0_{d\times d}$, the residual embedding matrices capture different aspects of the podcast in their feature columns, so the brain encoding model's weights reflect non-overlapping contributions from statistically independent features. In other words, each residual embedding matrix contributes to neural prediction via a sequence of samples that are uncorrelated with that of other residual matrices, enabling a clean separation of lexical, syntactic, meaning, and reasoning signals.
>
> **Why Are Section 3.1 and 3.2 complementary**
>
> Podcast-level uncorrelatedness (Section 3.1) and per-token geometric separation (Section 3.2) answer the same overarching question at two levels. Together, they provide a coherent picture of disentangled factorization: residuals add new (non-reused) signal globally, and that signal also aligns with orthogonal directions per token, yielding near-orthogonal feature subspaces across the podcast.
>
> We will make this explicit in the camera ready version: Section 3.1 establishes across-sample (podcast-level) orthogonality of feature columns, while Section 3.2 tests the stronger, per-token, feature-space near-orthogonality via cosine similarity.

---

### Official Review · Reviewer_vX2N · 2025-07-07

**Clarity:** 3
**Significance:** 2
**Originality:** 3
**Rating:** 5
**Confidence:** 4

**Summary:**

This paper proposes a residual‐disentanglement framework that isolates orthogonal LLM embeddings for lexicon, syntax, meaning, and reasoning, and uses the reasoning‐specific residuals to build encoding models that predict ECoG responses during natural speech.

**Questions:**

see weakness

**Ethical Concerns:**

["NO or VERY MINOR ethics concerns only"]

**Final Justification:**

Good for nips, I maintain acceptance.

**Limitations:**

see weakness

**Quality:**

3

**Strengths And Weaknesses:**

Strengths:
The idea is easy to follow and the paper is well written. The method clearly separates four linguistic feature embeddings and validates their independence using cosine‐similarity analysis.
It shows that the reasoning embedding captures unique variance in neural responses, peaks later, and engages higher‐level cortical areas.
It provides both a mathematical proof of approximate orthogonality and empirical evidence that the residual embeddings are independent.

Weaknesses:
The experiments use only ECoG data, which undersamples frontal regions and may limit spatial generalizability.
The reasoning probes rely only on COMPS language tasks without widely used advanced reasoning tasks like math and coding.
The study focuses only on Qwen2.5-14B, larger or differently trained models (reasoning models like r1 and qwq) may show different layering or reasoning features.

---

> ### Author Rebuttal · Authors · 2025-07-31
>
> ## Response to Reviewer vX2N
> First of all, we sincerely thank you for reviewing our paper and providing constructive comments! The followings specifically respond to key concerns you raised in your review.
>
> **Concern 1: Reasoning probes rely only on COMPS**
>
> **Summary of Response:** While our original analysis relied solely on COMPS-WUGS-DIST for identifying reasoning saturation layers, we have now added 5-hop deductive reasoning probes from ProntoQA to strengthen the evaluation. We applied both tasks to 17 Qwen models and found that the saturation layers they identify are closely aligned, with an average difference of only 0.94. This cross-validation suggests that our reasoning probe is robust across different task formats.
>
> **Full Response:** Thank you for highlighting this important point. In the original submission, we relied solely on COMPS-WUGS-DIST—a 1-hop deductive reasoning task—to identify the reasoning saturation layer, defined as the point at which reasoning performance plateaus within the model. While this task is useful for detecting the presence of reasoning capability, we acknowledge that its simplicity may limit generalizability to more complex forms of reasoning.
>
> Motivated by your feedback, we have incorporated a more challenging 5-hop deductive reasoning task from ProntoQA as an additional probe [1]. This task, which involves multi-step reasoning chains, has been widely adopted in recent reasoning trainings and benchmarks [2, 3, 4, 5], and we believe it serves as a meaningful enhancement to our evaluation framework.
>
> We applied both COMPS-WUGS-DIST and ProntoQA to 17 Qwen models spanning different sizes and versions. The results from the two tasks are highly consistent, with an average difference in reasoning saturation layer of only 0.94—small relative to the overall model depths (ranging from 25 to 49 layers). This consistency supports the robustness of our reasoning probe across tasks of varying complexity.
>
> | Model                        | ProntoQA | COMPS-WUGS-DIST |
> |------------------------------|----------|-----------------|
> | Qwen-1.8B                   | 13       | 14              |
> | Qwen-7B                     | 13       | 15              |
> | Qwen-14B                    | 18       | 20              |
> | Qwen1.5-1.8B                 | 13       | 14              |
> | Qwen1.5-7B                   | 14       | 16              |
> | Qwen1.5-14B                  | 20       | 20              |
> | Qwen2-1.5B                   | 17       | 17              |
> | Qwen2-7B                     | 16       | 18              |
> | Qwen2.5-1.5B                 | 16       | 18              |
> | Qwen2.5-7B                   | 16       | 19              |
> | Qwen2.5-14B                  | 28       | 28              |
> | Qwen3-1.7B-thinking-mode-off | 19       | 19              |
> | Qwen3-8B-thinking-mode-off   | 23       | 23              |
> | Qwen3-14B-thinking-mode-off  | 27       | 27              |
> | Qwen3-1.7B-thinking-mode-on  | 16       | 16              |
> | Qwen3-8B-thinking-mode-on    | 20       | 20              |
> | Qwen3-14B-thinking-mode-on   | 20       | 21              |
>
> **Concern 2: Only Qwen2.5-14B**
>
> **Summary of Response:** We have extended our probing analysis to a wide range of Qwen models beyond Qwen2.5-14B, revealing that while the precise saturation layers vary, all models consistently follow the same emergence order of syntax → meaning → reasoning. We also introduce a notion of relative depth to quantify how much of the model is dedicated to each feature. Our results show a clear trend across model generations: newer models allocate fewer layers to syntax and more to meaning and reasoning, suggesting a shift toward deeper processing of high-level features as model capabilities improve.
>
> **Full Response:** Thank you for raising this important question. You are absolutely right that probing across multiple models is essential to understanding whether feature emergence patterns generalize. Models with different architectures or training corpora may indeed exhibit different internal dynamics. In response, we applied our probing pipeline to a broad range of Qwen models. The results are summarized in the following table. While the precise saturation layers vary, we observe a consistent emergence order across all models—syntax → meaning → reasoning—with Qwen-1.8B being the only exception, where syntax and meaning saturate at the same layer.
>
> | Model                        | Syntax | Meaning | Reasoning |
> |------------------------------|--------|---------|-----------|
> | Qwen-1.8B                    | 11     | 11      | 14        |
> | Qwen-7B                      | 11     | 13      | 15        |
> | Qwen-14B                     | 9      | 16      | 20        |
> | Qwen1.5-1.8B                 | 10     | 11      | 14        |
> | Qwen1.5-7B                   | 9      | 13      | 16        |
> | Qwen1.5-14B                  | 8      | 16      | 20        |
> | Qwen2-1.5B                   | 10     | 14      | 17        |
> | Qwen2-7B                     | 7      | 14      | 18        |
> | Qwen2.5-1.5B                 | 7      | 14      | 18        |
> | Qwen2.5-7B                   | 7      | 15      | 19        |
> | Qwen2.5-14B                  | 6      | 20      | 28        |
> | Qwen3-1.7B-thinking-mode-off | 9      | 16      | 19        |
> | Qwen3-8B-thinking-mode-off   | 7      | 20      | 23        |
> | Qwen3-14B-thinking-mode-off  | 5      | 17      | 27        |
> | Qwen3-1.7B-thinking-mode-on  | 5      | 13      | 16        |
> | Qwen3-8B-thinking-mode-on    | 6      | 13      | 20        |
> | Qwen3-14B-thinking-mode-on   | 4      | 14      | 21        |
>
> Further, we analyzed broader trends emerging from these results. To do so, we introduce the notion of relative depth, which quantifies the proportion of layers dedicated to encoding each feature. Specifically, we define the relative depth of a feature as the fraction of total layers between the saturation layer of the preceding feature and that of the current one. For instance, in Qwen2.5-14B (which has 48 layers), meaning saturates at layer 20 and reasoning at layer 28, implying that layers 21–28 are primarily devoted to reasoning. Thus, the reasoning relative depth is computed as $(28 - 20) / 48$. More formally, for any feature $x \in \{s, m, r\}$ representing syntax, meaning, or reasoning, we define:
> $$
> Depth_x = \frac{L_x - L_{prev}}{n}
> $$
> where $L_x$ is the saturation layer of feature $x$, $L_{prev}$ is the saturation layer of the preceding feature, and $n$ is the total number of layers in the model. The table below reports these values for several 14B-scale models from the Qwen family.
>
> | Model                       | Syntax Depth | Meaning Depth | Reasoning Depth |
> |-----------------------------|--------------|---------------|-----------------|
> | Qwen-14B                    | 0.225        | 0.175         | 0.1             |
> | Qwen1.5-14B                 | 0.2          | 0.2           | 0.1             |
> | Qwen2.5-14B                 | 0.125        | 0.291         | 0.167           |
> | Qwen3-14B-thinking-mode-off | 0.125        | 0.3           | 0.25            |
> | Qwen3-14B-thinking-mode-on  | 0.1          | 0.25          | 0.175           |
>
> We observe a clear trend: as models evolve and likely improve in capability, they dedicate a smaller fraction of layers to encoding syntax, while allocating more capacity to meaning and reasoning. This suggests that newer models may encode low-level linguistic features more efficiently, allowing them to devote more layers to higher-level semantic and reasoning processes—potentially contributing to improved performance on complex tasks.
>
> **Concern 3: Only ECoG data**
>
> We appreciate your feedback. Indeed, assessing how the residual reasoning embedding aligns with brain signals beyond ECoG—such as fMRI or EEG—would be highly valuable, as these modalities offer complementary spatial coverage and signal characteristics. Among the available options, we selected the dataset that best matches our reasoning-focused research question while providing a good balance of spatial and temporal precision.
>
> Our experimental results show that reasoning differs from semantic and syntactic processing within a time window of about 100 ms—a timescale that is difficult to capture with fMRI. Regarding your point on limited frontal coverage, we note that while we have relatively good coverage over the inferior frontal gyrus (IFG), we do lack coverage in the prefrontal cortex, which is traditionally associated with higher-order cognition. This limitation arises because, in clinical settings, electrode placement for epilepsy surgery typically prioritizes coverage over the temporal lobe, where seizure foci are more common.
>
>
> **_We have updated our local draft accordingly and will incorporate these changes into the camera-ready version._**
>
>
> **References**
>
> [1] Abulhair Saparov and He He. Language Models are Greedy Reasoners: A Systematic Formal Analysis of Chain-of-Thought. In International Conference on Learning Representations (ICLR), 2023.
>
> [2] Shibo Hao, Yi Gu, Haodi Ma, Joshua Jiahua Hong, Zhen Wang, Daisy Zhe Wang, and Zhiting Hu. Reasoning with Language Model is Planning with World Model. arXiv preprint arXiv:2305.14992, 2023.
>
> [3] Pan, L., Albalak, A., Wang, X., & Wang, W. Y. (2023). LOGIC-LM: Empowering Large Language Models with Symbolic Solvers for Faithful Logical Reasoning.
>
> [4] Tian, X., Ji, Y., Wang, H., Chen, S., Zhao, S., Peng, Y., Zhao, H., & Li, X. (2025). Not All Correct Answers Are Equal: Why Your Distillation Source Matters. arXiv preprint arXiv:2505.14464v2.
>
> [5] Xu, J., Fei, H., Pan, L., Liu, Q., Lee, M.-L., & Hsu, W. (2024). Faithful Logical Reasoning via Symbolic Chain-of-Thought. arXiv preprint arXiv:2405.18357v2.

---

> ### Comment · Area_Chair_sDnY · 2025-08-04
> **Subject: Friendly Reminder to Acknowledge or Update Your Review**
>
> Dear Reviewer vX2N,
>
> Thank you for your time and effort in reviewing the submissions and for providing valuable feedback to the authors.
>
> If you haven’t already done so, we kindly remind you to review the authors’ rebuttals and respond accordingly. In particular, if your evaluation of the paper has changed, please update your review and explain the revision. If not, we would appreciate it if you could acknowledge the rebuttal by clicking the “Rebuttal Acknowledgement” button at your earliest convenience.
>
> This step ensures smooth communication and helps us move forward efficiently with the review process.
>
> We sincerely appreciate your dedication and collaboration.
>
> Best regards,
> AC

---

> ### Comment · Area_Chair_sDnY · 2025-08-06
>
> Dear Reviewer vX2N,
>
>
>
> The authors have provided responses to your questions. Could you please review their reply to see whether your concerns have been adequately addressed, and let us know if you have any further questions or comments?
>
> Please also refer to the following message from the Program Chairs:
>
> "Reviewers are expected to stay engaged in discussions, initiate them and respond to authors’ rebuttal, ask questions and listen to answers to help clarify remaining issues. It is not OK to stay quiet. It is not OK to leave discussions till the last moment. If authors have resolved your (rebuttal) questions, do tell them so. If authors have not resolved your (rebuttal) questions, do tell them so too."
>
> Afterward, please kindly submit your Reviewer Acknowledgement as well.
>
> Best regards,
>
> AC

---

### Note · Authors · 2025-08-16

We sincerely thank all reviewers and our Area Char (AC) for their insightful feedback and active engagement. We appreciate the reviewers' recognise of our work!

**Contributions & strengths**
Our paper introduces a residual disentanglement pipeline that (i) localizes layers where syntax/meaning/reasoning saturate via minimal-pair probes, (ii) regresses out lower-level signals to form near-independent lexicon/syntax/meaning/reasoning features, and (iii) uses them for ECoG encoding. This reveals unique predictive power for reasoning residuals, later/frontal spatiotemporal dynamics, and clarifies encoding results using raw high-layer states can be biased by shallow cues. The work ties theory, diagnostics (cosine matrices, probing on residuals, variance partitioning), and neural evidence into a coherent, interpretable mapping.

**Reviewer comments & our responses**
1. Validity across LLM architectures.
   We expanded probing to 17 Qwen models (1.8B–14B). A consistent syntax → meaning → reasoning emergence order appears across scales.

2. Generalization to other reasoning tasks.
   We added a 5-hop deductive probe (ProntoQA) and, per suggestion, WinoGrande. Across all 17 models, reasoning saturation layers from COMPS-WUGS-DIST, ProntoQA, and WinoGrande closely align (avg differences 0.94 and 0.53 layers), supporting robustness of the probing pipeline and reasoning saturation layer.

3. Do residuals capture the intended constructs?
   In addition to mathematical rationale and cosine-similarity analyses, we also evaluated each residual embedding (syntax/meaning/reasoning) on all three probing tasks, yielding a 3×3 matrix: each residual embedding performs best on its own task and significantly worse on others, demonstrating feature specificity and effective disentanglement.

4. Brain encoding with additional models.
   We tested Qwen-14B and Qwen2.5-1.5B alongside Qwen2.5-14B, enabling comparisons across generation and scale. Stronger reasoning ability and newer generations correspond to higher brain–model correlations for reasoning residuals.

5. Neuroscientific grounding of reasoning-related regions.
   We added a table mapping regions from our spatiotemporal analysis to prior literature on reasoning/cognitive control, confirming alignment with well-documented networks.

All comments have been resolved, with corresponding analyses, figures, and textual clarifications merged into the current draft. The manuscript now reflects these updates throughout.

---

### Decision · Program_Chairs · 2025-09-17

**Decision:**

Accept (poster)

**Comment:**

This paper presents a residual disentanglement approach for isolating four separate linguistic feature embeddings (lexical, syntactic, semantic, and reasoning) from LLMs by detecting saturation layers via probing and progressively eliminating lower-level features. Subsequently, the paper employs the reasoning-specific residuals to construct encoding models that predict ECoG responses during natural speech processing.

This work proposes a novel and intuitive residual disentanglement framework that isolates four linguistically meaningful feature embeddings (lexical, syntactic, semantic, and reasoning) from LLMs. The approach is theoretically grounded through a mutual independence theorem that guarantees feature separation, while empirical results demonstrate validation—near-zero cosine similarities between residual embeddings provide evidence for successful disentanglement.

However, one reviewer acknowledges the paper's promising idea but maintains concerns about unsubstantiated claims regarding LLM-derived reasoning embeddings specifically mapping to human brain patterns. Thus, the authors are strongly suggested to show the regions/signals detected by the reasoning embeddings indeed take center stage in the human reasoning process, and show the other embeddings struggle to identify these same regions.